# Modelling Analysis of a Four-Way Shuttle-Based Storage and Retrieval System on the Basis of Operation Strategy

Jia Mao [1], Jinyuan Cheng [1], Xiangyu Li [2], Honggang Zhao [3,*] and Ciyun Lin [1]

1 School of Transportation, Jilin University, Changchun 130022, China
2 College of Automotive Engineering, Jilin University, Changchun 130022, China
3 College of Chemical Engineering, Xinjiang Normal University, Urumqi 830054, China
* Correspondence: 107621997010002@xjnu.edu.cn

**Abstract:** In the context of sustainable development, this paper rationalises the outbound process of a four-way shuttle system with a focus on their modelling, performance evaluation and configuration using a parallel operation strategy to reduce resource waste, thus achieving sustainable development. The parallelism of the hoist and shuttle is innovatively incorporated into the four-way shuttle system, so the modelling content is divided into parallel and serial types. In the parallel operation strategy model, a separation–aggregation queueing network model is constructed, and the open-loop queueing network is innovatively solved using the maximum entropy method. In the serial operation strategy model, a semi-open-loop queuing network is constructed and solved using the geometric matrix method. By varying different parameters, the accuracy of the model is verified by Arena simulation with an error range of 10% or less, and the error of the system performance index calculation is reduced by 20% compared with the existing methods. Setting up 18 different sizes of shuttle systems provided a better performance than a single serial-operation strategy through the addition of parallel strategies, with an average reduction of 12.6% in the system response time and a minimum reduction of 1.8%. The conclusions of this paper were verified on the basis of an arithmetic case analysis.

**Keywords:** automated warehouses; four-way shuttle systems; queuing network model; analytical and numerical modelling; performance analysis

## 1. Introduction

With the rapid development of information technology and e-commerce, green transportation, green consumption and other areas of high-quality development, green low-carbon lifestyle and production techniques are accelerating the adoption of the concept of sustainable development among the people. As a third source of profit, the logistics industry has gained unprecedented momentum with respect to development. Warehouse systems are the material and information transit centers of enterprise logistics systems and have a decisive influence on the operational efficiency and cost of the whole supply chain [1]. Four-way shuttle systems are the most advanced system currently available, and represent a highly automated, highly flexible and dense storage system, the appearance of which has improved the performance of storage systems, avoiding the defects of traditional automated warehouse equipment, which typically cover a large area, and possess low operational efficiency and poor flexibility. However, the lack of a good operating strategy will result in operational inefficiency, wasted resources and other problems, which are in opposition to the concept of sustainable development. For this reason, it is of great theoretical and practical significance to establish a theoretical model for a four-way shuttle system that is in line with reality, and to conduct research on the operation strategy and system performance. The existing scholarly research on queuing networks in storage systems is shown in Table 1.

**Table 1.** Summary of research on AVS/RS systems.

| Author | Warehousing System | Model Type | Modeling | Solving Method |
|---|---|---|---|---|
| Kuo et al. [2] | Multilayer AVS/RS | System performance analysis | NQN | Iterative method |
| Fukunari and Malmborg [3] | Multilayer AVS/RS | System performance analysis | OQN | Meanvalue analysis |
| Fukunari and Malmborg (2009) [4], Kuo et al. [5] | Multilayer AVS/RS | System performance analysis | CQN | Meanvalue analysis |
| Heragu et al. [6] | Monolayer AVS/RS | System performance and configuration analysis | OQN | Manufacturing system performance analyser |
| Ekren et al. [7] | Multilayer AVS/RS | Statistic analysis | SOQN | Simulation method |
| Ekren et al. [8] | Multilayer AVS/RS | System performance analysis | Simulation model | Simulation method |
| Guerrazzi et al. [9] | Multilayer AVS/RS | System performance analysis | Energy Model | Simulation method |
| Roy D et al. [10–14] | Monolayer AVS/RS | System performance analysis | SOQN | Decomposition method Iterative method, Decomposition method |
| Antonio et al. [15] | Multilayer AVS/RS | System performance analysis | Evaluation System Model | Decomposition method |
| Ekren et al. [16] | Multilayer AVS/RS | System performance analysis | Simulation Regression model | Decomposition method |

For the study of dense storage systems, the establishment of a system model is an essential optimisation step. Scholars have mostly used the queuing network model; however, for complex shuttle systems, a single open-loop or closed-loop queuing network cannot meet the modelling needs under multilayer, multishuttle and multilift conditions. Therefore, it is also necessary to study the system's performance in order to establish a model that reflects the actual operation mode of the system. At present, studies of storage system shuttle and hoist operation strategies mostly use a single serial strategy; however, parallel operation strategies have been employed in real scenarios, and have a significant positive impact on the system performance. AVS/RS systems, AS/RS systems and SBS/RS systems have been developed to analyse the impact of the operating strategy on the operational efficiency of the whole system [17–19]. It can be observed from the available literature that dense storage systems have been widely studied by scholars, and the research has mostly been centred on AVS/RS. The research directions of some scholars have mostly focused on system scheduling, storage strategy and sorting order optimisation, while other scholars have focused on system modelling, system performance evaluation and system configuration. In addition to there being less research on four-way shuttle systems, the research has the following limitations: (1) it mostly presents single operation strategies for the shuttle and hoist in the system; (2) it mostly presents single-level and single-aisle analyses; and (3) generality is ignored when building queuing networks [20–22].

Therefore, this study focuses on a four-way shuttle system with a parallel operation strategy; adopts system performance indices, such as equipment utilisation, system response time and external queue waiting time; establishes a separation–aggregation queuing network model and a SOQN model; and designs an algorithm based on the Coxian distribution and the MGM solution. The performance variation of the system is analysed under different operation strategies [23–26].

## 2. Problem Description

This paper studies a single-depth four-way shuttle system. The hoist can assist the four-way shuttle in achieving layer changes, but it can also carry goods to complete the entry and exit operations. Figure 1 shows a simple diagram, in which the shuttle car and layer change hoist can be configured according to the specific situation. Usually, one hoist serves multiple lanes and multiple shuttles within the system, and the shuttles move within the corresponding area. The possible movements of the four-way shuttle car are: straight ahead, lane change and layer change; straight-ahead and lane-change movements are performed by themselves, while layer-change movements are performed with the assistance of the layer change hoist.

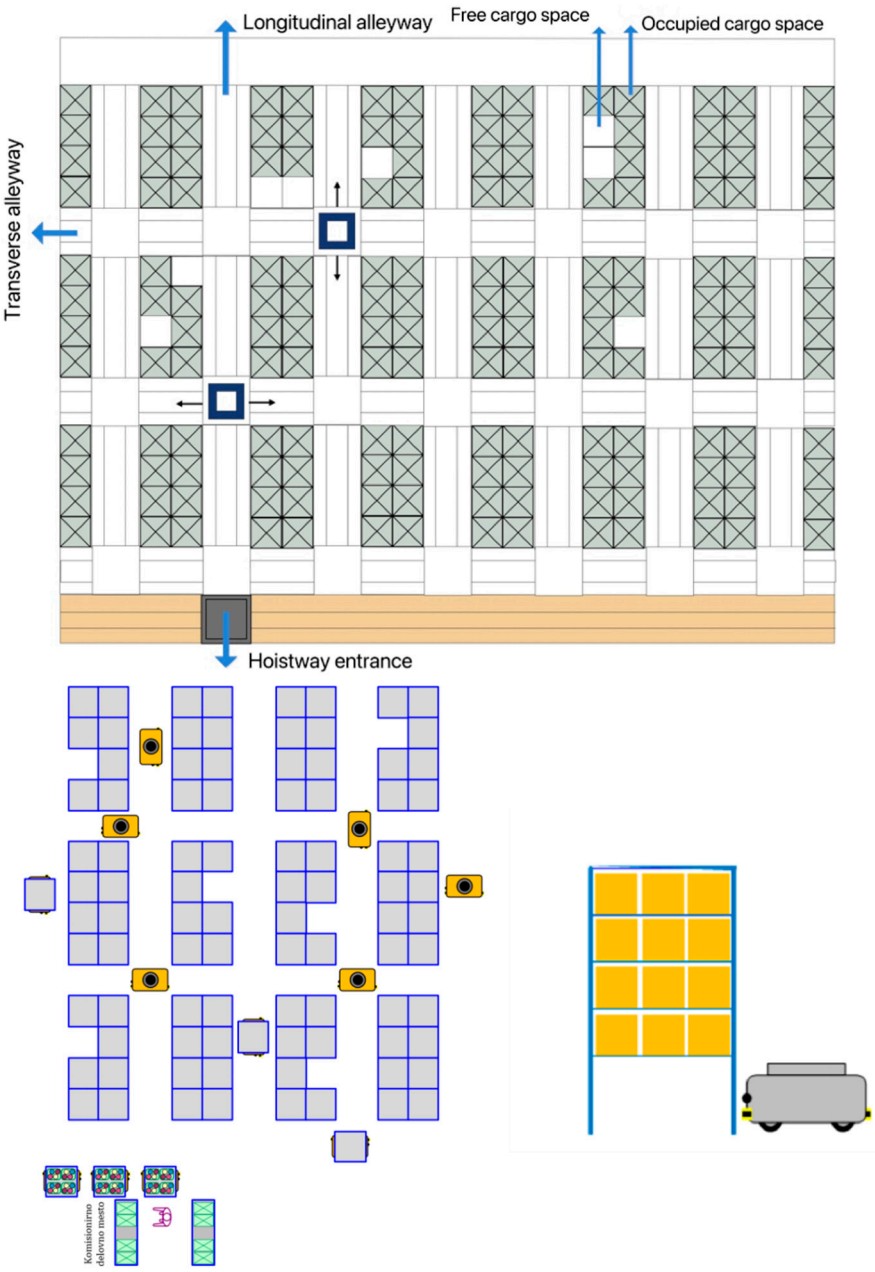

**Figure 1.** Schematic diagram of a four-way shuttle system.

The basic operation modes of the four-way shuttle system can be divided into inbound and outbound operations. As the inbound and outbound operations are logically similar,

and the steps involved in the outbound operations include those of the inbound operations, this paper focuses on the outbound operations with the following flow:

In the outbound operation process, the FCFS strategy is used, and the WMS locates the layer, column and lane of the SKU according to the outbound tasks [27–30].

### 3. System Outbound Process Modelling

*3.1. System Outbound Process Modelling Assumptions*

To establish a reasonable model, the assumptions and unknown parameters first need to be established. First, the object of study in this paper is a four-way shuttle system with one hoist; the aisles it serves, as well as the shuttles it serves, consist of an overall multilayer, multi-aisle unit for mapping the performance of the whole system. The system uses one tote as the access unit, and only one tote can be accessed per bay, with the same bay size. For the sake of convenience, and without loss of generality, the following assumptions are made [31]:

1. The shuttles and hoists obey the FCFS strategy.
2. The main reason for studying the outbound process is that both the inbound and outbound operations are logically coordinated with the shuttle and hoist, while the outbound operation involves steps that encompass the inbound operation.
3. The goods follow a random storage strategy, and the tasks are assigned using a random assignment strategy in which each unused space has the same probability of being used to store goods, and each shuttle and hoist has the same probability of being assigned to a task.
4. The hoist and the shuttle follow the POSC (endpoint stay) strategy, which means that when the shuttle and the hoist complete a certain task, they stay at the end position of the task until the next task arrives. In this paper, the hoist will stay at the I/O location (system level 1 location) when it completes the pickup task, and the shuttle will stay in the cache area of the level on which the previous pickup task was located.
5. The operation time of the shuttle and hoist obeys a general distribution.
6. A single-cycle operation is adopted.

With assumption 5, it is established in this paper that the service time of the shuttle and hoist obey a general distribution, and the queuing network composed of the general distribution takes the form of a nonproduct solution; therefore, the general model for solving queuing networks is not suitable for use in this paper. However, some scholars have studied the approximate relationship between the general distribution and the Coxian-type distribution in storage systems (see [32]), which can be used to simplify the calculation. The Coxian distribution belongs to a special form of PH constructed from a series of exponential distributions, which can be applied in the approximate solution of a service time while obeying the general distribution [33].

*3.2. Average Service Time Model for a Four-Way Shuttle System Server*

To describe the four-way shuttle system based on the operation time model in detail, the parameters used in the model and their meanings are expressed as shown in Table 2.

**Table 2.** Meaning of unknown parameters in the model.

| Parameters | Meaning | Parameters | Meaning |
|---|---|---|---|
| $T$ | Storage levels of shelves | $W_A$ | Roadway width (m) |
| $C$ | storage columns for shelves | $H$ | Single storage rack height (m) |
| $N$ | Number of tunnels served by a single hoist | $Vs$ | The average speed of shuttle (m/s) |
| $A$ | Number of longitudinal aisles of single-level shelves | $\varepsilon_s$ | Average loading/unloading time for shuttles (t) |
| $S$ | Total storage system capacity | $\gamma_s$ | Time delay caused by shuttle acceleration/deceleration (t) |
| $L$ | Length of individual cargo space (m) | $Vl$ | The average speed of hoist (m/s) |
| $W$ | Width of individual cargo space (m) | $\varepsilon_l$ | Average loading/unloading time of hoist (t) |
| $M$ | Number of shuttles served by a single hoist | $\gamma_l$ | Time delay due to hoist acceleration/deceleration (t) |

Set the pickup task at the t tier as a customer of class t. From assumption 3, the probability of serving a customer of class t is:

$$P(t) = \frac{1}{T} \tag{1}$$

Set the goods to be picked up in column $M_{ac}$ of aisle t, where $a = 1, 2, \ldots, N$, $c = 1, 2, \ldots, C$; then, the probability of picking up the goods is:

$$P(M_{ac}) = \frac{1}{2 \cdot N \cdot C} \tag{2}$$

The calculation of the equipment service time can be divided into two cases, depending on the availability of the idle four-way shuttles in the layer of goods to be picked up. These are the parallel operation strategy, where there are free four-way shuttles in the pickup layer, and the serial operation strategy, where there are no free four-way shuttles in the pickup layer.

3.2.1. Server Service Time Model Using Parallel Operation Strategy

First, the shuttle and the hoist need to carry out horizontal pickup operations and vertical target-level seating operations, respectively, constituting a parallel operation strategy in which the shuttle and the hoist complete their tasks at the same time. Subsequently, the shuttle carries the goods to the hoist, the hoist accepts the goods to complete the convergence, and finally, the shuttle carries the goods to level 1. The probability in this case $P_1$ is:

$$P_1 = \frac{C_{T-1}^{M-1}}{C_T^M} \tag{3}$$

Therefore, the average service time of the server in vertical jobs, horizontal jobs and remaining jobs are calculated as follows:

7. Operation service time of vertical operations

In accordance with assumption 4, this paper adopts the endpoint stay strategy. The hoist stays at the endpoint of the previous task, which is the initial position of system level 1; therefore, the movement distance of the vertical job is the vertical distance from the target level to level 1. Thus, the service time of vertical operation $T_{l1}(t)$ is:

$$T_{l1}(t) = \frac{(t-1)H}{V_t} + 2(\varepsilon_l + \gamma_l) \tag{4}$$

8. Operating service time of horizontal operations

When the four-way shuttle is at the target t level, the movement distance can be divided into (1) moving from the hoistway entrance (buffer zone) to the target cargo location; and (2) returning to the hoistway entrance. The distance of movement includes the horizontal distance across the lane and the vertical distance along with the lane movement, which involves the cross-aisle movement of the shuttle. According to expert consultation, the number of cross-aisles $x$ is determined by the number of aisles served by hoist $N$. If $N$ is an even number, the location of the hoist is the $\frac{N}{2}$th aisle from left to right; if $N$ is an odd number, the location of the hoist is the $\frac{N+1}{2}$ th aisle from left to right, then the number of aisles $x$ spanned takes a range of values $0 \le x \le \left\lfloor \frac{N}{2} \right\rfloor$ and the probability of crossing aisle $x$ is:

$$P(x) = \begin{cases} \frac{1}{N}, & x = 0 \\ \frac{1}{N}, & N \text{ is even and } x = \frac{2}{N} \\ \frac{2}{N}, & N \text{ is even and } 0 < x < \left\lfloor \frac{N}{2} \right\rfloor \\ \frac{2}{N}, & N \text{ is odd and } 0 < x < \left\lfloor \frac{N}{2} \right\rfloor \end{cases} \tag{5}$$

The value of $x$ is:

$$x = \left|\left|\left\lceil \frac{N}{2} \right\rceil - a\right|\right| \tag{6}$$

Then, the service time of the shuttle during this phase $T_S(M_{ac})$ is:

$$T_s(M_{ac}) = \frac{2\left(x + \left\lceil \frac{c}{4} \right\rceil\right)W_A + 2Lx + cW}{V_s} + \varepsilon_s + 4\gamma_s \tag{7}$$

9.    Operating service time of the remaining operations

The hoist moves from the target t level with cargo to level 1, and the operating service time $T_{ls}(t)$ is:

$$T_{ls}(t) = \frac{(t-1)H}{V_l} + 2(\varepsilon_l + \gamma_l) \tag{8}$$

In this case, there is a parallel operation between the hoist movement in 1 and the shuttle movement in 2. While the shuttle receives the outbound task, it applies for the hoist to wait at the target level; if one side arrives first, it will wait for the other side until it completes the rendezvous at the hoistway entrance picking up the goods and unloading them with the hoist to level 1.

3.2.2. Service Time Model for the Server Using a Serial Operation Strategy

In this case, the hoist needs to carry out three operations: the first is to move to the layer with the idle shuttle, to pick up the trolley; the second is for the idle four-way shuttle to move to the target cargo layer when the hoist that is still in the task queue is not released; and the third is for the hoist to carry the goods to the first layer to complete the pickup task; at this time, the shuttle and the hoist are released. The probability in this case $P_2$ is:

$$P_2 = 1 - P_1 \tag{9}$$

Then, the average service times of the server for vertical jobs, horizontal jobs and remaining jobs are calculated as follows:

1.    Operation service time for vertical operations

The hoist is initially at level 1 under assumption 4, and the idle four-way shuttle is located at the position it was in at the end of the last job, which is in the cache of level $t'$, on which the last job was located. Therefore, the service time of the movement of the hoist to the idle four-way shuttle $T_l(It')$ is:

$$T_l(It') = \frac{(t-1)H}{V_l} + 2\gamma_l \tag{10}$$

The pickup target level is $t$, and the operating service time for moving the idle shuttle on the hoist to the target level $T_{ls}(t't)$ is:

$$T_{ls}(t't) = \frac{|t' - t|H}{V_l} + 2(\varepsilon_l + \gamma_l) \tag{11}$$

The total vertical operation time $T_l(t')$ is:

$$T_l(t') = \frac{|t' - t| + (t-1)H}{V_l} + 2(2\gamma_l + \varepsilon_l) \tag{12}$$

2.    Operating service time of horizontal operations

Similar to the case described in Section 4.2.1, the cross-aisle probability is:

$$P_1(x) = P(x) \tag{13}$$

Therefore, the service time of the shuttle car to complete the horizontal operation is:

$$T_{s2}(M_{ac}) = T_s(M_{ac}) \tag{14}$$

3.  The operating service time of the remaining operations is:

$$T_{ls}(t') = T_{ls}(t) \tag{15}$$

Due to assumptions 4 and 6, the shuttle and hoist are not released until they complete a job; thus, no parallelism occurs in this case.

### 3.3. Modelling the Outbound Process of a Four-Way Shuttle System Based on Two Operation Strategies

The parallel and serial queuing network model for the four-way shuttle system outbound process are shown in Figure 2.

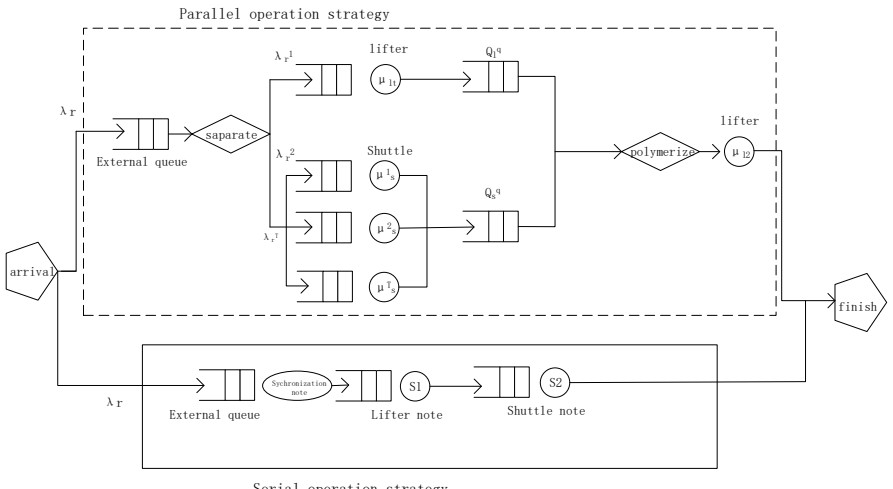

**Figure 2.** Four-way shuttle system outbound process queuing network model.

### 3.3.1. Parallel Part Modelling

Construct the separation–aggregation queuing network model and analyse the parallel operation strategy by decomposing the complex network method in the following steps:

1.  Constructing the separation–aggregation queuing network model

In this section, we have shown that the whole parallel operation strategy is divided into vertical operations, horizontal operations and remaining operations. First, the pickup order arrives at the "separation" point at arrival rate $\lambda_r$, at which time the pickup task is divided into a vertical operation dominated by a hoist and a horizontal operation dominated by a shuttle, where the $T$ shelves are divided into T different types of customers, arriving at arrival rate $\lambda_r^1, \lambda_r^2, \ldots, \lambda_r^T$. Next, the hoist and the shuttle wait in the vertical queue $Q_l$ and the horizontal queue $Q_s$, respectively. After both the vertical and horizontal operations are completed, they converge at the "convergence" point; finally, the hoist carries the goods to complete the last remaining operations. As assumption 6 obeys the single-command cycle, the lifters are not released until the task is completed; therefore, the lifters under node $\mu_{l2}$ can provide an unlimited service capacity. The separation aggregation queuing network model is shown in Figure 3.

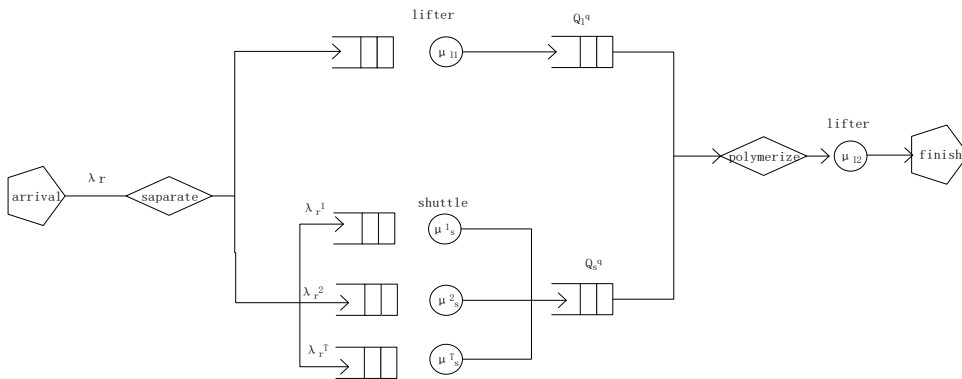

**Figure 3.** Separation-aggregation queuing network model.

2. Constructing a closed-loop queueing network

As the shuttle and hoist obey a general distribution, the parallel operation strategy described in Section 3.3.1 is a nonproduct solution and cannot be solved directly; thus, the Coxian distribution solution model, which approximates the general distribution, is applied in this paper. Figure 4 shows a schematic diagram of a closed-loop queuing network under a parallel strategy, where there are $K_r$ pick-up operations in the closed-loop network and the arrival rate of each tier of jobs is $\lambda_r^i = \frac{\lambda_r}{T}$. After entering this network, the shuttles are already bound to the order, the maximum number of accesses that can be accepted in the system cannot exceed the number of shuttles and the time used in the jobs to receive service at the shuttles and hoists is equal to their own service time.

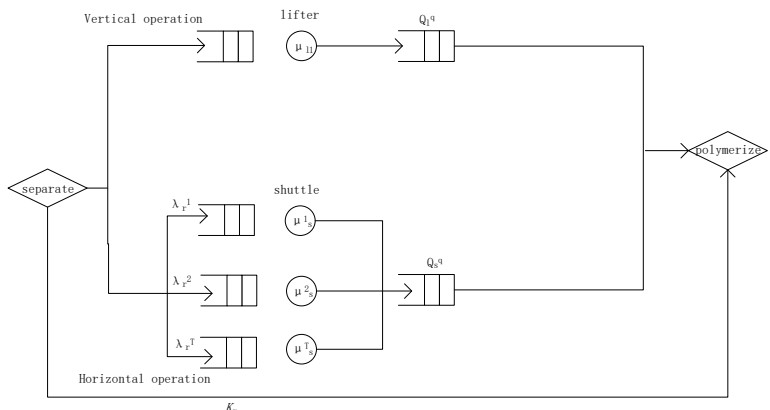

**Figure 4.** Closed-loop queuing network using the parallel strategy.

The importance of building this closed-loop queuing network model is to create an equivalent service node to the open-loop queuing network model, which can simplify the complex queuing network.

3. Constructing an open-loop queuing network

First, the service point formed by the closed-loop queuing network is denoted as node 1, and the service rate of this service point obeys a negative exponential distribution, which leads to an open-loop queuing network with two servers in series, where the service rate of service point 1 is replaced by the throughput rate sought in 2, which is load dependent. In the open-loop queuing network, the pickup task arrives with an arrival rate of $\lambda_r$, node 1 enters node 2 for the remaining operations after equivalence and as node 1 is composed of a closed-loop queuing network, it compensates for the lack of trolley waiting time for the orders in the open-loop network. Finally, the maximum entropy method is used to solve the metrics of this queuing network model. The open-loop queuing network formed under the parallel strategy is shown in Figure 5.

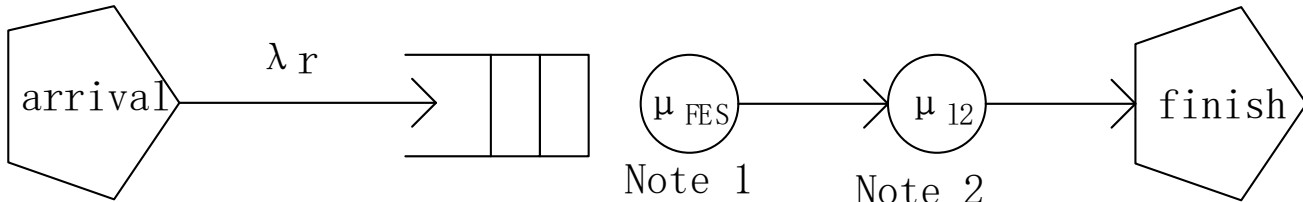

**Figure 5.** Open-loop queuing network using the parallel strategy.

3.3.2. Serial Part Modelling

If there are no idle shuttles at the target level, the serial operation strategy will be employed, constituting a semi-open-loop queuing network, as shown in Figure 6. The "synchronisation node" can be understood as the system receiving the pickup command. The pickup command, after entering the system and the shuttle for binding, is entered as a task in the queue of the waiting hoist node S1; if there is an idle hoist, it can complete the task, whereby a hoist picks up an idle shuttle. After completing the task at shuttle node S2, which is waiting to pick up the goods, the hoist picks up the shuttle and moves it to the target layer to pick up the goods; throughout the whole process, the shuttle is still in the task and is not released; therefore, the shuttle node is able to directly carry out the task of picking up the goods until the hoist receives the goods once they have been released. Finally, we consider hoist node S3, where the remaining operations are performed. Similar to node S2, the hoist is not released during the task, and therefore, any remaining operations do not need to wait.

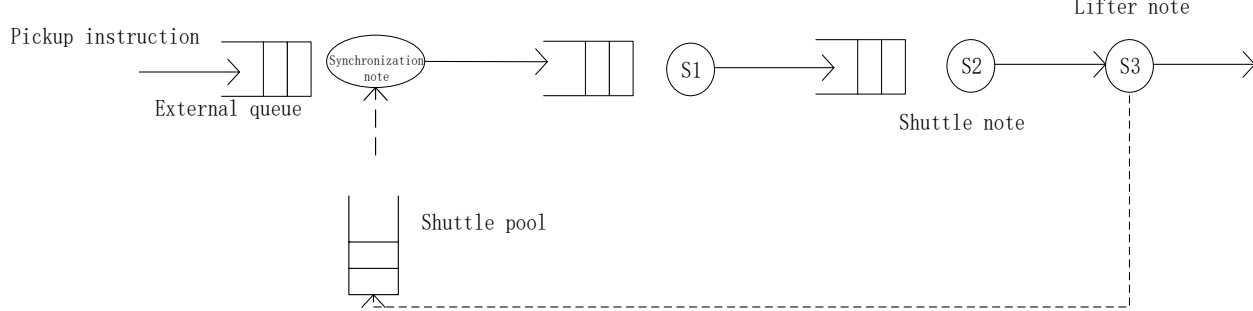

**Figure 6.** Semi-open-loop queuing network using the serial strategy.

Similar to the parallel operation strategy, the Coxian distribution is used to approximate the general distribution of the service times instead. In the process of solving the semi-open-loop queuing network model, the use of more than two service nodes will cause a sharp increase in the state space, meaning that the model cannot be computed. Therefore, two nodes need to be aggregated into one, and as hoist node S3 does not need to be queued for direct service, hoist nodes S1 and S3 are combined into one, as shown in Figure 7.

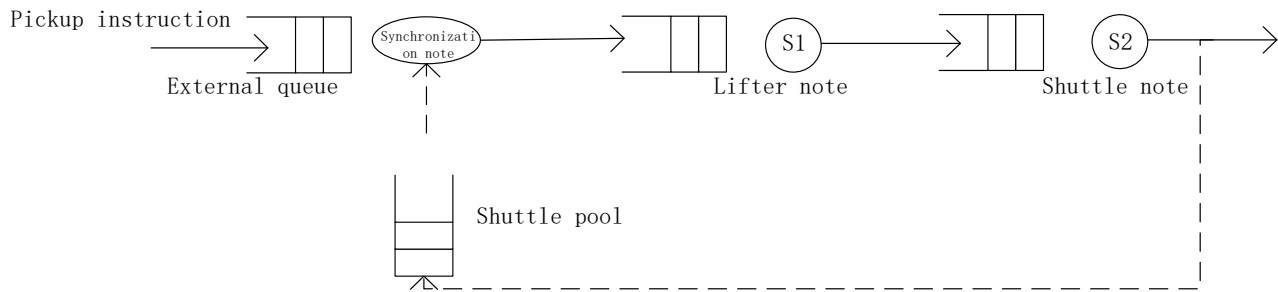

**Figure 7.** Integrated semi-open-loop queueing network.

The whole system model is composed based on the probabilities of the two cases under the completed parallel operation strategy and the serial operation strategy.

## 4. Model Solution

### 4.1. Solving the Separation-Aggregation Queueing Network Model Using the Parallel Strategy

As the operating times for the horizontal operations, vertical operations and remaining operations obey the general distribution, and the Coxian distribution is used as an approximation of this in this paper, the mean value of the horizontal operating time can be derived from Equations (2) and (7):

$$\tau_s = \sum_{a=1}^{N} \sum_{c=1}^{C} P(M_{ac}) T_s(M_{ac}) \tag{16}$$

where $a = 1, 2 \ldots N$, $c = 1, 2, \ldots, C$ and the coefficient of the variation of the square of the horizontal operating time is $cv_s^2$. Similarly, the mean values of the vertical operation and remaining operation time $\tau_{l1}, \tau_{l2}$, can be expressed according to Equations (1), (4) and (8), as follows:

$$\tau_{l1} = \sum_{t=1}^{T} P(t) T_{l1}(t) \tag{17}$$

$$\tau_{l2} = \sum_{t=1}^{T} P(t) T_{ls}(t) \tag{18}$$

Additionally, set the squared coefficients of the variation for the vertical operation and the remaining operation times are set as $cv_{l1}^2$, $cv_{l2}^2$.

Thus, the service rates of the horizontal operation of the shuttle, the vertical operation of the hoist and the remaining operation of the hoist, $\mu_s$, $\mu_{l1}$ and $\mu_{l2}$ can be expressed by their mean values, as follows:

$$\mu_s = \frac{1}{\tau_s}, t = 1, 2, \ldots, T \tag{19a}$$

$$\mu_{l1} = \frac{1}{\tau_{l1}} \tag{19b}$$

$$\mu_{l2} = \frac{1}{\tau_{l2}} \tag{19c}$$

#### 4.1.1. Constructing a Closed-Loop Queuing Network to Compute State-Based Service Rates

A closed-loop queuing network model based on separation-aggregation is shown in Figure 2, where there are $K_r$ pickups in the closed-loop network, $Q_l$ is the number of customers waiting to be served in $Q_l^q$ and $Q_s D$ is the number of customers waiting to be served in $Q_s^q$. The state variable $s_k$ of this closed-loop queueing network can be given as:

$$s_k = (Q_s, Q_L), k = Q_l + Q_s + mQ_l = Q_l(m+1) + Q_s \tag{20}$$

where $m = \min(K_r, M)$ is the maximum number of customers waiting to be served in $Q_s^q$ and $Q_l + Q_s < m + 1$, $Q_l = 0, 1$, $Q_s = 0, 1, \ldots, m$.

If the system completes an aggregation using the parallel operation strategy, there are two scenarios:

1. At $Q_l = 1$, the hoist completes its operation and reaches the convergence point. This process requires the shuttle to complete the pickup operation and move to the entrance of the aisle; then, the service rate of this process is $\mu_s$.

2. At $Q_s > 1$, the shuttle completes the pickup operation and arrives at the hoistway entrance, and this process requires the hoistway to move to the entrance of the target level. Then, the service rate of this process is $\mu_{l1}$.

Setting $\pi(s_k)$ as the state probability in state $s_k$, the throughput rate can be expressed by the two cases of aggregation as:

$$\lambda_f(K_r) = \sum_{Q_s=0}^{m} \mu_{l1}\pi(Q_s,0) + \sum_{Q_s=0}^{m-1} \mu_s\pi(Q_s,1) \tag{21}$$

The service time of shuttle service point $\mu_s$ is independently and identically distributed, denoted as $(\alpha,\ T)$, where the initial probability vector $\alpha = \begin{bmatrix} 1 & 0 & 0 & 0 & \ldots \end{bmatrix}_{1\times p}$ is the initial probability vector of the p-order Coxian distribution and T is the p-order square matrix. The service time of hoist service point $\mu_{l1}$ is independently and identically distributed, denoted as $(\beta,\ S)$, and the initial probability vector $\beta = \begin{bmatrix} 1 & 0 & 0 & 0 & \ldots \end{bmatrix}_{1\times p}$ is the initial probability vector of the q-order Coxian distribution and S is the q-order square matrix.

The state variables of the separation-aggregation closed-loop queuing network model are set to $(s_k, i, j)$, where $i, j$ are the states in which the shuttle and hoist are located, respectively, where $i = 1, 2, 3 \ldots p;\ and\ j = 1, 2, 3 \ldots q$. Let $\pi_t$ be the steady-state probability distribution of $s_k$. As the two cases complete the parallel operation strategy complete the aggregation, $\pi_t = [\pi_0, \pi_1]$.

Therefore, the state transfer rate matrix Q of the state variable $s_k$ is:

$$Q = \begin{bmatrix} B_{00} & B_{01} \\ B_{10} & B \end{bmatrix} \tag{22}$$

The transfer diagram of the generation and extinction process is shown in Figure 8.

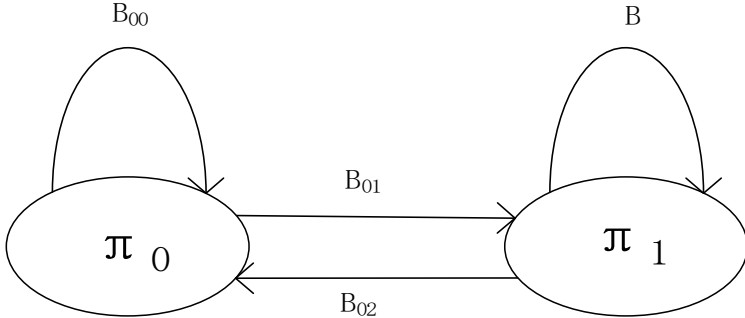

**Figure 8.** Transfer diagram of the generation and extinction process.

$B_{01}$ represents the rate matrix for the transfer of the $\pi_0 = [\pi(s_0), \pi(s_1), \pi(s_2) \ldots \pi(s_m)]$ state to the $\pi_1 = \left[ \pi\left(s_{m+1)}, \pi(s_{m+2}), \pi(s_{m+1}), \pi(s_{m+3}), \ldots, \pi(s_{2m}) \right) \right]$ state.

$$B_{01} = \begin{bmatrix} (Q_s, Q_l) & (0,1) & (1,1) & (2,1) & \ldots & (m-1,1) \\ (0,0) & I_T \otimes S^\circ & & & & \\ (1,0) & & I_T \otimes S^\circ & & & \\ (2,0) & & & I_T \otimes S^\circ & & \\ \vdots & & & & \ddots & \\ (m-1,0) & & & & & I_T \otimes S^\circ \\ (m,0) & & & & & 0 \end{bmatrix} \tag{23}$$

$B_{00}$ represents the rate matrix for the transfer of the $\pi_0 = [\pi(s_0), \pi(s_1), \pi(s_2) \ldots \pi(s_m)]$ state to the $\pi_0 = [\pi(s_0), \pi(s_1), \pi(s_2) \ldots \pi(s_m)]$ state.

$$
B_{00} =
\begin{bmatrix}
(Q_s, Q_l) & (0,0) & (1,0) & (2,0) & (2,1) & \ldots & (m-1,0) & (m,0) \\
(0,0) & T \oplus S & T\alpha \otimes I_S & & & & & \\
(1,0) & I_T \otimes S\beta & T \oplus S - I_T \otimes \widetilde{S}^\circ & T^\circ \alpha \otimes I_S & & & & \\
(2,0) & & I_T \otimes S\beta & T \oplus S - I_T \otimes \widetilde{S}^\circ & T^\circ \alpha \otimes I_S & & & \\
\vdots & & & & \ddots & & & \\
(m-1,0) & & & & & I_T \otimes S\beta & T \oplus S - I_T \otimes \widetilde{S}^\circ & T^\circ \otimes I_S \\
(m,0) & & & & & & S^\circ \beta \otimes \alpha & S
\end{bmatrix}
\tag{24}
$$

where $I_T$ and $I_S$ are unit matrices that have the same order as $T$ and $S$, respectively, $\otimes$ is the Kronecker product, $\oplus$ is the Kronecker sum and $T \oplus S = T \otimes I_S + I_T \otimes S$. In the state transfer rate matrix, the sum of all the elements in each row is 0, so that $\widetilde{S}^\circ$ is found to be:

$$
\widetilde{S}^\circ =
\begin{bmatrix}
\mu_2^\beta \left(1 - b_1^\beta\right) \\
\mu_1^\beta \left(1 - b_2^\beta\right) \\
\vdots \\
\mu_{q-1}^\beta \left(1 - b_{q-1}^\beta\right) \\
\mu_q^\beta
\end{bmatrix}
\tag{25}
$$

$B_{10}$ is the rate matrix of the transfer from the $\pi_1 = [\pi(s_0), \pi(s_1), \pi(s_2) \ldots \pi(s_m)]$ state to the $\pi_1 = [\pi(s_0), \pi(s_1), \pi(s_2) \ldots \pi(s_m)]$ state.

$$
B_{01} =
\begin{bmatrix}
(Q_s, Q_l) & (0,0) & (1,1) & (2,1) & \ldots & (m-1,0) & (m,0) \\
(0,1) & T^\circ \alpha & & & & & \\
(1,1) & & T^\circ \alpha \otimes \beta & & & & \\
(2,1) & & & T^\circ \alpha \otimes \beta & & & \\
\vdots & & & & \ddots & & \\
(m-1,1) & & & & & T^\circ \alpha \otimes \beta & 0
\end{bmatrix}
\tag{26}
$$

$B$ is the rate matrix of the transfer from the $\pi_1 = [\pi(s_0), \pi(s_1), \pi(s_2) \ldots \pi(s_m)]$ state to the $\pi_1 = [\pi(s_0), \pi(s_1), \pi(s_2) \ldots \pi(s_m)]$ state.

$$
B =
\begin{bmatrix}
(Q_s, Q_l) & (0,1) & (1,1) & (2,1) & \ldots & (m-1,1) \\
(0,1) & T - \widetilde{T}^\circ & T^\circ \alpha & & & \\
(1,1) & & T - \widetilde{T}^\circ & T^\circ \alpha & & \\
(2,1) & & & T - \widetilde{T}^\circ & & \\
\vdots & & & & \ddots & T^\circ \alpha \\
(m-1,1) & & & & & T - \widetilde{T}^\circ
\end{bmatrix}
\tag{27}
$$

The $\widetilde{T}^\circ$ is as follows:

$$
\widetilde{T}^\circ =
\begin{bmatrix}
\mu_1^\alpha \left(1 - b_1^\alpha\right) \\
\mu_2^\alpha \left(1 - b_2^\alpha\right) \\
\vdots \\
\mu_{P-1}^\beta \left(1 - b_{p-1}^\beta\right) \\
\mu_p^\alpha
\end{bmatrix}
\tag{28}
$$

3. According to the smooth equation $\begin{cases} \vec{\pi} \cdot Q = \vec{0} \\ \vec{\pi} \cdot e = 1 \end{cases}$ , the equilibrium equation of the model is obtained as follows:

$$\begin{bmatrix} \pi_0 & \pi_1 \end{bmatrix} \cdot Q = \vec{0} \begin{bmatrix} \pi_0 & \pi_1 \end{bmatrix} \cdot e = 1 \tag{29}$$

### 4.1.2. An Open-Loop Queueing Network Is Constructed to Calculate the Separation-Aggregation Queueing Network Model Metrics

After the calculations in the previous subsection, the system can reduce the subnetwork in the separation-aggregation queuing network to an equivalent simultaneous flow of servers, with the hoist forming a serial open-loop queuing network, which assumes a negative exponential distribution for node 1, whose service rate is determined by the throughput rate and the load in the system; thus, it is set to $\mu_{FES}(x) = \lambda_f(x)$.

Setting the state probability in the open-loop queuing network as $\pi(k_1, k_2)$, $k_1$ and $k_2$ as the number of customers served by node 1 and node 2 and $\pi(k_1)$, $\pi(k_2)$ as the approximate probabilities of node 1 and node 2, respectively. The state probability of this network can be given by the number of customers served by each node.

$$\pi(k_1, k_2) = \pi_1(k_1) \cdot \pi_2(k_2) \tag{30}$$

(1) For service point 1, the average service rate and the average number of customers served are $\pi_1(k_1)$; this node can be found on the basis of the approximate probability $\overline{K}_1$. The equilibrium equation is shown in Table 3, and the birth and death process is shown in Figure 9.

**Table 3.** Equilibrium equation transfer table.

| Number of Customers (Status) | Equilibrium Equation | Simplified Equations |
|---|---|---|
| 0 | $\pi_1(0)\lambda_r = \pi_1(1)\mu_{FES}(1)$ | $\pi_1(1) = \pi_1(0)\frac{\lambda_r}{\mu_{FES}(1)}$ |
| 1 | $\pi_1(1)(\mu_{FES}(1) + \lambda_r) = \pi_1(0)\lambda_r + \pi_1(2)\mu_{FES}(2)$ | $\pi_1(2) = \pi_1(0)\frac{\lambda_r^2}{\mu_{FES}(1) \cdot \mu_{FES}(2)}$ |
| 2 | $\pi_1(2)(\mu_{FES}(2) + \lambda_r) = \pi_1(1)\lambda_r + \pi_1(3)\mu_{FES}(3)$ | $\pi_1(3) = \pi_1(0)\frac{\lambda_r^2}{\mu_{FES}(1) \cdot \mu_{FES}(2) \cdot}\mu_{FES}(1)$ |
| ... | ... | ... |

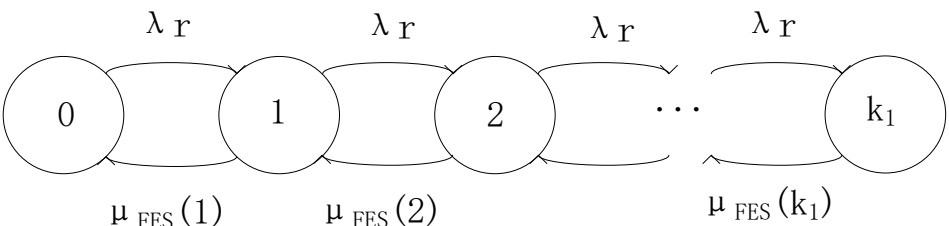

**Figure 9.** Extinction process.

The approximate probability of finding node 1 is:

$$\pi_1(0) = \frac{1}{1 + \sum_{k_1}^{+\infty} \prod_{i=1}^{k_1} \frac{\lambda_r}{\mu_{FES}(i)}} \tag{31a}$$

$$\pi_1(k_1) = \pi_1(0) \prod_{i=1}^{k_1} \frac{\lambda_r}{\mu_{FES}(i)} \tag{31b}$$

As the service rate of node 1 is equivalently represented by the throughput rate calculated in Section 4.1.1 and the throughput rate is related to the load of node 1, the average service rate of node 1 is:

$$\mu_1 = \sum_{k_1}^{+\infty} \pi(k_1) \cdot \mu_{FES}(k_1) \tag{32}$$

The average number of customers served at node 1, $\overline{K}_1$, can be calculated from the approximate probability:

$$\overline{K}_1 = \sum_{k_1}^{+\infty} k_1 \cdot \pi_1(k_1) \tag{33}$$

(2) Solving node 2

As node 1 obeys the negative exponential distribution and node 2 obeys the general distribution, the queueing network formed is a nonproduct solution and the queueing network formed is not easy to solve; therefore, in this paper, the MEM is applied to the solution of the open-loop queueing network, where the entropy is the measure of uncertainty that can be predicted for something. For node 2, the entropy function is introduced according to the maximum entropy principle in the form of a universal ME solution $\pi_2(n)$ by maximising the entropy generalisation function $H(p)$ under the following constraints [34,35]:

$$H(\pi) = -\sum_{n=1}^{k_2} \pi_2(n) \ln[\pi_2(n)], k_2 \geq 1 \tag{34}$$

The optimisation model of the maximum is:

$$\max H(\pi) = -\sum_{n=1}^{k_2} \pi_2(n) \ln[\pi_2(n)] \tag{35}$$

$$\text{s.t.} \begin{cases} \sum_{n=0}^{k_2} p(n) = 1 \\ \sum_{n=1}^{k_2} h(n)\pi_2(n) = \rho \\ \sum_{k=1}^{k_2} n\pi_2(n) = L \end{cases} \tag{36}$$

The approximate probability of node 2 is:

$$\pi_2(n) = \begin{cases} (1-\rho) & n=0 \\ (1-\rho)gx^n & n \geq 1 \end{cases} \tag{37}$$

where $g = \frac{\rho_2^2}{(1-\rho_2)(L-\rho_2)}, x = \frac{L-\rho_2}{L}$ and $\rho_2 = \frac{\lambda_r}{\mu_{l2}}$, applying the generalised Laplace equation and the z-transform equation [36,37], showing that for any queue, the average queue length is given by the same equation:

$$L = \frac{\rho_2}{2}\left(1 + \frac{C_{a_2}^2 + \rho_2 C_s^2}{1-\rho_2}\right) \tag{38}$$

where $C_{a_2}^2$ is the squared coefficient of the variation of the customer arrival time at node 2 and $C_s^2$ is the squared coefficient of the variation of the service time at node 2 [38].

Setting the average number of customers at node 2 to $\overline{K}_2$, the approximate probability can be derived from:

$$\overline{K}_2 = \sum_{k_2=1}^{+\infty} k_2 \cdot \pi_2(k_2) \tag{39}$$

(3)   Solving for system performance metrics

1. The system response time, which is the time required to complete a pickup operation, including the waiting time and the service time, can be derived using Little's theorem.

$$E[T] = \frac{\overline{K}_1}{\lambda_r} + \frac{1}{\mu_{l2}} \tag{40}$$

2. The waiting time in the external queue for pickup is:

$$W_r = \frac{\overline{K}_1}{\lambda_r} - \frac{1}{\overline{\mu}_1} \tag{41}$$

3. The external queue waiting for pickup is:

$$L_{eq} = W_r \cdot \lambda_r \tag{42}$$

4. As this paper studies multiple shuttles in a cell, the arrival rate of each shuttle is $\frac{\lambda_r}{M}$ and the utilisation rate of the shuttles can be calculated as:

$$\rho_s = \frac{\lambda_r}{M \cdot \mu_s} \tag{43}$$

*4.2. Solving the Semi-Open-Loop Queuing Network Model under a Serial Strategy*

4.2.1. Solving the Transfer Rate Matrix Q

Define the system state as $(n_1, n_2, i, j, k)$, where $i \geq 0$ is the number of external queue pickup orders, $0 \leq j \leq M$ is the number of hoist node pickup orders and $0 \leq k \leq M$ is the number of shuttle node pickup orders. Set $n_1 = i + j$ as the sum of the number of the external queues and first-node pickup orders for the sake of the solution convenience of solving and combine it with $n_2 = k$ to form the birth and death process of the two states.

If the sum of the external queue, the first node and the second node pickup order is greater than M, then there must be idle four-way shuttles in the system, and the external queue waiting for pickup orders will be 0. If the sum of the external queue, the first node and the second node pickup order number is more than M, then there are no idle shuttles in the system, and the whole system is busy, so the external queue is defined as [39]:

$$i = \begin{cases} 0 & n_1 + n_2 \leq M \\ n_1 + n_2 - M & n_1 + n_2 > M \end{cases} \tag{44}$$

When the performance metrics are calculated (e.g., average queue length), only $n_1$ and $n_2$ are relevant. Therefore, for the sake of brevity, this semi-open-loop intraqueueing network state variable is modified to $(n_1, n_2)$. Defining the steady-state probability vector as $\pi_0 = \left( \pi_{(0,0)}, \pi_{(0,1)}, \pi_{(0,2)} \dots \right), \pi_0 = \left( \pi_{(i,0)}, \pi_{(i,1)}, \pi_{(i,2)} \dots \right) \pi_i = \left( \pi_{(i,0)}, \pi_{(i,1)}, \pi_{(i,2)} \dots \right)$, on the basis of the Coxian distribution, where $(\eta, H)$ denotes the arrival interval time distribution, $(\omega, W)$ denotes the service time of node 1 and $(\gamma., U)$ denotes the service time of node 2, the transfer rate matrix can be expressed as:

$$Q = \begin{bmatrix} B_0 & C_0 & & \\ A_1 & B & C & \\ & A & B & C \\ & \ddots & \ddots & \ddots \end{bmatrix} \tag{45}$$

where $B_0$ is the rate matrix for the transfer from the $\pi_0 = \left(\pi_{(0,0)}, \pi_{(0,1)}, \pi_{(0,2)} \dots\right)$ state to the $\pi_0 = \left(\pi_{(0,0)}, \pi_{(0,1)}, \pi_{(0,2)} \dots\right)$ state, which is an $(M+1) \times (M+1)$-squared matrix, M is the number of shuttles served by the hoist, $I^H$ is a unit matrix of the same order as H, $U^\circ$ is the absorption probability matrix of U, $\otimes$ is the Kronecker product and $\oplus$ is the Kronecker sum.

A transfer diagram of the generation and extinction process is shown in Figure 10.

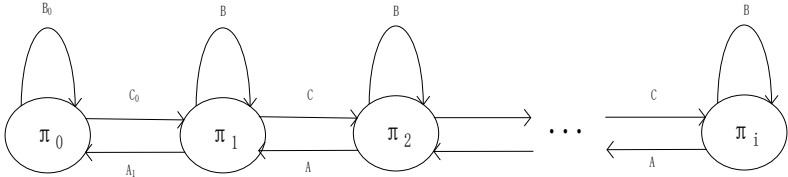

**Figure 10.** Transfer diagram of the generation and extinction process.

$$
B_0 = \begin{bmatrix}
(n_1, n_2) & (0,0) & (0,1) & (0,2) & \dots & (0,M) \\
(0,0) & H & & & & \\
(0,1) & I^H \otimes U^\circ & H \oplus U & & & \\
(0,2) & & I^H \otimes U^\circ \gamma & H \oplus U & & \\
\vdots & & \ddots & & \ddots & \\
(0,M) & & & & I^H \otimes U^\circ \gamma & H \oplus U
\end{bmatrix}
\tag{46}
$$

$C_0$ is the rate matrix of the transfer from the $\pi_0 = \left(\pi_{(0,0)}, \pi_{(0,1)}, \pi_{(0,2)} \dots\right)$ state to the $\pi_1 = \left(\pi_{(1,0)}, \pi_{(1,1)}, \pi_{(1,2)} \dots\right)$ state, $I^U$ is the unit matrix of the same order as U, and $H^\circ$ is the absorption probability matrix of H.

$$
C_0 = \begin{bmatrix}
(n_1, n_2) & (1,0) & (1,1) & (1,2) & \dots & (1,M) \\
(0,0) & H^\circ \eta \otimes \omega & & & & \\
(0,1) & & H^\circ \eta \otimes \omega \otimes I^U & & & \\
(0,2) & & & H^\circ \eta \otimes \omega \otimes I^U & & \\
\vdots & & & & \ddots & \\
(0,M) & & & & & H^\circ \eta \otimes I^U
\end{bmatrix}
\tag{47}
$$

$A_1$ is the rate matrix of the transfer from that state to the $\pi_0 = \left(\pi_{(0,0)}, \pi_{(0,1)}, \pi_{(0,2)} \dots\right)$ state and $W^\circ$ is the absorption probability matrix of the same order as $W$. As the upper limit of the capacity of node 2 is $M$, the transfer rate is 0 in the transfer from the state $(1, M)$ to $(0, M)$.

$$
A_1 = \begin{bmatrix}
(n_1, n_2) & (0,0) & (0,1) & (0,2) & \dots & (0,M) \\
(1,0) & & I^H \otimes W^\circ \otimes \gamma & & & \\
(1,1) & & & I^H \otimes W^\circ \otimes I^U & & \\
\vdots & & & & \ddots & \\
(1, M-1) & & & & & I^H \otimes W^\circ \otimes I^U \\
(1, M) & & & & & 0
\end{bmatrix}
\tag{48}
$$

$B$ is the rate matrix of the $\pi_1 = \left(\pi_{(1,0)}, \pi_{(1,1)}, \pi_{(1,2)} \dots\right)$ to $\pi_1 = \left(\pi_{(1,0)}, \pi_{(1,1)}, \pi_{(1,2)} \dots\right)$ state transfer, and $I^W$ represents the unit matrix of the same order as $W$.

$$
B = \begin{bmatrix}
(n_1, n_2) & (1,0) & (1,1) & \cdots & (1, M-1) & (1, M) \\
(1,0) & H \oplus W & & & & \\
(1,1) & I^H \otimes I^W \otimes U^\circ & H \oplus W \otimes U & & & \\
\vdots & & & \ddots & & \\
(1, M-1) & & & & I^H \otimes I^W \otimes U^\circ \gamma & H \oplus W \oplus U \\
(1, M) & & & & I^H \otimes \omega \otimes U^\circ \gamma & H \oplus U
\end{bmatrix}
\tag{49}
$$

C is the rate matrix for the transfer from the $\pi_1 = \left( \pi_{(1,0)}, \pi_{(1,1)}, \pi_{(1,2)} \dots \right)$ to the $\pi_2 = \left( \pi_{(2,0)}, \pi_{(2,1)}, \pi_{(2,2)} \dots \right)$ state and $A$ is the rate matrix for the transfer from the $\pi_2 = \left( \pi_{(2,0)}, \pi_{(2,1)}, \pi_{(2,2)} \dots \right)$ to the $\pi_1 = \left( \pi_{(1,0)}, \pi_{(1,1)}, \pi_{(1,2)} \dots \right)$ state.

$$
C = \begin{bmatrix}
(n_1, n_2) & (1,0) & (2,1) & \cdots & (2, M-1) & (2, M) \\
(1,0) & H^\circ \eta \otimes I^W & & & & \\
(1,1) & & H^\circ \eta \otimes I^W \otimes I^U & & & \\
\vdots & & & \ddots & & \\
(1, M-1) & & & & H^\circ \eta \otimes I^W \otimes I^U & \\
(1, M) & & & & & H^\circ \eta \otimes I^U
\end{bmatrix}
\tag{50}
$$

$$
A = \begin{bmatrix}
(n_1, n_2) & (1,0) & (1,1) & (1,2) & \cdots & (2, M) \\
(2,0) & & I^H \otimes W^\circ \otimes \gamma & & & \\
(2,1) & & & I^H \otimes W^\circ \otimes I^U & & \\
\vdots & & & & \ddots & \\
(2, M-1) & & & & & I^H \otimes W^\circ \otimes I^U \\
(2, M) & & & & & 0
\end{bmatrix}
\tag{51}
$$

### 4.2.2. Solving for the Steady-State Probability Distribution

If a Markov state exists as a steady-state, then $\pi Q = 0$, using the steady-state probability vector $\pi_0 = \left( \pi_{(0,0)}, \pi_{(0,1)}, \pi_{(0,2)} \dots \right)$, $\pi_i = \left( \pi_{(i,0)}, \pi_{(i,1)}, \pi_{(i,2)} \dots \right)$, rewriting $\pi Q = 0$ as a matrix equation yields [39,40]:

$$
\begin{aligned}
\pi_0 B + \pi_1 A_1 &= 0 \\
\pi_0 C_0 + \pi_1 B + \pi_2 A &= 0 \\
\pi_1 C + \pi_2 B + \pi_3 A &= 0 \\
&\vdots \\
\pi_{i-1} C + \pi_i B + \pi_{i+1} A &= 0
\end{aligned}
\tag{52}
$$

where $\pi_0, \pi_i$ is a row vector of $1 \times (M + 1)$, $\pi_i$ is denoted as the steady-state probability vector of $i$ in each state and 0 is a row vector with element 0. The class of the birth and death state can be classed as a constant return state, i.e., it returns to itself with a probability of 1 in a finite number of steps, so the relationship between the steady-state probabilities can be written as:

$$
\pi_i = \pi_0 R^i
\tag{53a}
$$

$$
\pi_{i+1} = \pi_i R
\tag{53b}
$$

where $R$ is the probability matrix as a square matrix of $(M + 1) \times (M + 1)$; therefore, the formula can be simplified as:

$$
C + RB + R^2 A = 0 \Rightarrow R = -\left( C + R^2 A \right) B^{-1}
\tag{54}
$$

4.2.3. Solving the Performance Index

After solving the steady-state probability vectors, the captain of each node can be derived from the expected value of each state. Determine the state space of shuttle node $S_2$ as $S_2 = [0, 1, 2, 3, \ldots, M, 0, 1, 2, 3, \ldots.M, 0, 1, \ldots]$; then, the captain $L_{S_2}$ of shuttle node $S_2$ is:

$$L_{S_2} = \pi_0 \begin{bmatrix} 0 \\ 1 \\ 2 \\ \vdots \\ M \end{bmatrix} + \pi_1 \begin{bmatrix} 0 \\ 1 \\ 2 \\ \vdots \\ M \end{bmatrix} + \ldots + \pi_i \begin{bmatrix} 0 \\ 1 \\ 2 \\ \vdots \\ M \end{bmatrix} + \ldots \tag{55}$$

Then, the shuttle utilisation rate $\rho_s$ is:

$$\rho_s = \frac{L_{S_2}}{M} \tag{56}$$

The total captain $L_S$ of the external queue and elevator node $S_1$ is:

$$L_S = \sum_{i=1}^{\infty} \pi_1 R^{i-1} ie = \pi_1 (I - R)^{-1} e \tag{57}$$

The external queue captain $L_{eq}$ is [41]:

$$L_{eq} = \sum_{i=1}^{\infty} |\pi_i| \tag{58}$$

where $|\pi_i|$ is the sum of the elements in $\pi_i$ and the external queue waiting time $W_r$ is:

$$W_r = \frac{L_{eq}}{\lambda_r} \tag{59}$$

Then, the captain $L_{S1}$ of the hoist node is:

$$L_{S1} = L_S - L_{eq} \tag{60}$$

The system response time $E(T)$ for the nodal captain of the hoist and the shuttle as is shown in Equation (61).

$$E(T) = \frac{L_{S1} + L_{S2}}{\lambda_r} \tag{61}$$

## 5. Simulation Test and Data Analysis

### 5.1. Arena Simulation Validation

As the scale of the four-way shuttle system varies according to the size of the business volume, this paper lists seven different four-way shuttle system scales with different numbers of layers and columns and sets five different arrival rates. The other assumptions of the system are consistent with the model established in this paper, i.e., only pickup operations are considered; the first come, first served principle is adopted; and the random storage strategy is adopted to verify the reliability and accuracy of the model at different scales and arrival rates. The reliability and accuracy of the model are verified for different sizes and arrival rates. The initial system parameters set in the simulation are shown in Table 4, the system size is shown in Table 5 and the arrival rate is set to $\lambda_r$ = 50, 100, 150, 200, 250 per hour.

**Table 4.** Simulation parameter values.

| Parameters | Take Value | Parameters | Take Value |
|---|---|---|---|
| $Vs$ (m/s) | 2 | $\gamma_l$ (s) | 2 |
| $\varepsilon_s$ (s) | 3 | $L$ (m) | 1.2 |
| $\gamma_s$ (s) | 2 | $W$ (m) | 1 |
| $Vl$ (m/s) | 3 | $WA$ (m) | 1 |
| $\varepsilon_l$ (s) | 1 | $H$ (m) | 1.5 |
| $N$ | 3 | $M$ | 7 |

**Table 5.** Simulation scale.

| Size | Small Scale | | | | Large Scale | | |
|---|---|---|---|---|---|---|---|
| Serial number | 1 | 2 | 3 | 4 | 5 | 6 | 7 |
| layers | 5 | 6 | 7 | 8 | 9 | 10 | 11 |
| columns | 36 | 40 | 44 | 48 | 64 | 72 | 80 |
| Capacities | 360 | 480 | 616 | 768 | 1152 | 1440 | 1760 |

　　　Before commencing the seven simulation scenarios, a warm-up period of 100 h was implemented to eliminate errors and stabilise the system, followed by a confidence interval of 95% and a simulation half-step of less than 2% of the mean value, which was run ten times, averaging over 2000 h each time, in order to obtain the final results. The metrics calculated for the simulation are the system response time, the external queue waiting for pickup and the shuttle utilisation. The error value between the simulation results and the model solution results was calculated using Equation (62), where $R_1$ represents the simulation results and $R_2$ represents the model solution results:

$$\xi = \frac{|R_1 - R_2|}{R_1} \times 100\% \tag{62}$$

　　　From the results in Table 6, it can be concluded that the $\varsigma_{E(T)}$ error average is 5.01%, the $\varsigma_{L_{eq}}$ error average is 4.58% and the $\varsigma_{\rho_S}$ error average is 2.11%. With the increasing arrival rate and system scale, the error percentage also increases, reaching a value of 16.72% because, while the number of shuttles in the simulation remains constant at five, with the increasing system scale and arrival rate, the shuttles are not able to meet the requirements of high-intensity system operation, thus generating higher error rates. However, as a whole, 74.32% of the data have an error range below 5%, which is sufficient to prove that the model in this paper is able to interpret the four-way shuttle system well; the error range is shown in Figure 11.

**Table 6.** Comparison of simulation results and theoretical values.

| Number | | 1 | | | 2 | | | 3 | | | 4 | | | 5 | | | 6 | |
|---|---|---|---|---|---|---|---|---|---|---|---|---|---|---|---|---|---|---|
| $\lambda_r = 50$ | $R_1$ | $R_2$ | $\zeta$ | $R_1$ | $R_2$ | $\zeta$ | $R_1$ | $R_2$ | $\zeta$ | $R_1$ | $R_2$ | $\zeta$ | $R_1$ | $R_2$ | $\zeta$ | $R_1$ | $R_2$ | $\zeta$ |
| $E(T)$ | 86.26 | 88.46 | 2.55 | 94.11 | 98.25 | 4.40 | 106.07 | 107.69 | 1.53 | 110.44 | 113.90 | 3.14 | 133.58 | 139.57 | 4.48 | 148.24 | 153.02 | 3.22 |
| $W_r$ | 37.03 | 37.40 | 0.99 | 62.42 | 60.18 | 3.59 | 67.98 | 64.70 | 4.82 | 69.71 | 66.25 | 4.96 | 90.12 | 87.11 | 3.34 | 100.01 | 95.91 | 4.10 |
| $L_{eq}$ | 0.51 | 0.52 | 0.99 | 0.87 | 0.84 | 3.59 | 0.94 | 0.90 | 4.82 | 0.97 | 0.92 | 4.96 | 1.25 | 1.21 | 3.34 | 1.39 | 1.33 | 4.10 |
| $\rho_s$ | 0.15 | 0.15 | 0.21 | 0.16 | 0.16 | 1.00 | 0.14 | 0.15 | 5.82 | 0.14 | 0.13 | 3.53 | 0.15 | 0.15 | 0.13 | 0.15 | 0.15 | 0.39 |
| $\lambda_r = 100$ | $R_1$ | $R_2$ | $\zeta$ | $R_1$ | $R_2$ | $\zeta$ | $R_1$ | $R_2$ | $\zeta$ | $R_1$ | $R_2$ | $\zeta$ | $R_1$ | $R_2$ | $\zeta$ | $R_1$ | $R_2$ | $\zeta$ |
| $E(T)$ | 95.27 | 96.78 | 1.58 | 102.78 | 106.94 | 4.05 | 110.25 | 114.73 | 4.06 | 114.18 | 110.59 | 3.15 | 140.14 | 145.26 | 3.65 | 154.77 | 166.82 | 7.79 |
| $W_r$ | 64.13 | 63.71 | 0.65 | 69.01 | 68.85 | 0.23 | 74.18 | 73.76 | 0.56 | 76.43 | 77.56 | 1.48 | 97.24 | 97.78 | 0.56 | 108.81 | 109.68 | 0.80 |
| $L_{eq}$ | 1.78 | 1.77 | 0.65 | 1.92 | 1.91 | 0.23 | 2.06 | 2.05 | 0.56 | 2.12 | 2.15 | 1.48 | 2.70 | 2.72 | 0.56 | 3.02 | 3.05 | 0.80 |
| $\rho_s$ | 0.34 | 0.35 | 2.94 | 0.32 | 0.32 | 0.94 | 0.29 | 0.30 | 4.21 | 0.28 | 0.27 | 2.07 | 0.32 | 0.30 | 5.17 | 0.30 | 0.30 | 1.53 |

**Table 6.** *Cont.*

| Number | 1 | | | 2 | | | 3 | | | 4 | | | 5 | | | 6 | | |
|---|---|---|---|---|---|---|---|---|---|---|---|---|---|---|---|---|---|---|
| $\lambda_r = 150$ | $R_1$ | $R_2$ | $\zeta$ | $R_1$ | $R_2$ | $\zeta$ | $R_1$ | $R_2$ | $\zeta$ | $R_1$ | $R_2$ | $\zeta$ | $R_1$ | $R_2$ | $\zeta$ | $R_1$ | $R_2$ | $\zeta$ |
| $E(T)$ | 110.37 | 105.89 | 4.06 | 117.37 | 116.50 | 0.74 | 123.70 | 124.53 | 0.67 | 126.45 | 124.69 | 1.39 | 153.86 | 163.77 | 6.44 | 174.07 | 182.07 | 4.60 |
| $W_r$ | 76.89 | 78.77 | 2.45 | 80.94 | 85.56 | 5.71 | 84.63 | 89.92 | 6.25 | 85.75 | 88.43 | 3.13 | 111.26 | 120.10 | 7.95 | 123.62 | 130.86 | 5.86 |
| $L_{eq}$ | 3.20 | 3.28 | 0.10 | 3.37 | 3.57 | 5.71 | 3.53 | 3.75 | 0.26 | 3.57 | 3.68 | 0.13 | 4.64 | 5.00 | 7.95 | 5.15 | 5.45 | 5.86 |
| $\rho_s$ | 0.51 | 0.53 | 2.35 | 0.48 | 0.48 | 0.39 | 0.45 | 0.45 | 0.69 | 0.41 | 0.42 | 1.46 | 0.45 | 0.46 | 0.44 | 0.45 | 0.46 | 0.88 |
| $\lambda_r = 200$ | $R_1$ | $R_2$ | $\zeta$ | $R_1$ | $R_2$ | $\zeta$ | $R_1$ | $R_2$ | $\zeta$ | $R_1$ | $R_2$ | $\zeta$ | $R_1$ | $R_2$ | $\zeta$ | $R_1$ | $R_2$ | $\zeta$ |
| $E(T)$ | 124.14 | 119.83 | 3.47 | 144.07 | 133.11 | 7.61 | 145.05 | 136.89 | 5.63 | 145.37 | 140.10 | 3.63 | 188.47 | 177.51 | 5.81 | 208.30 | 197.58 | 5.15 |
| $W_r$ | 92.91 | 90.63 | 2.45 | 107.31 | 107.39 | 0.07 | 107.20 | 106.40 | 0.75 | 102.74 | 105.45 | 2.64 | 139.47 | 141.07 | 1.15 | 155.13 | 153.68 | 0.94 |
| $L_{eq}$ | 5.16 | 5.04 | 0.14 | 5.96 | 5.97 | 0.07 | 5.96 | 5.91 | 0.04 | 5.71 | 5.86 | 0.15 | 7.75 | 7.84 | 1.15 | 8.62 | 8.54 | 0.94 |
| $\rho_s$ | 0.70 | 0.70 | 0.46 | 0.78 | 0.77 | 1.41 | 0.60 | 0.60 | 0.72 | 0.55 | 0.56 | 0.91 | 0.61 | 0.61 | 0.20 | 0.61 | 0.61 | 0.59 |
| $\lambda_r = 250$ | $R_1$ | $R_2$ | $\zeta$ | $R_1$ | $R_2$ | $\zeta$ | $R_1$ | $R_2$ | $\zeta$ | $R_1$ | $R_2$ | $\zeta$ | $R_1$ | $R_2$ | $\zeta$ | $R_1$ | $R_2$ | $\zeta$ |
| $E(T)$ | 128.23 | 127.45 | 0.61 | 172.63 | 153.42 | 11.13 | 164.92 | 146.27 | 11.31 | 170.04 | 152.16 | 10.51 | 215.88 | 190.03 | 11.98 | 254.21 | 211.70 | 16.72 |
| $W_r$ | 98.23 | 99.92 | 1.72 | 155.41 | 124.10 | 20.15 | 130.98 | 118.18 | 9.77 | 139.65 | 121.48 | 13.01 | 184.04 | 156.66 | 14.88 | 198.91 | 171.99 | 13.53 |
| $L_{eq}$ | 5.46 | 5.55 | 1.72 | 10.79 | 8.62 | 1.40 | 9.10 | 8.21 | 0.68 | 9.70 | 8.44 | 0.90 | 12.78 | 10.88 | 14.88 | 13.81 | 11.94 | 13.53 |
| $\rho_s$ | 0.87 | 0.88 | 0.58 | 0.80 | 0.81 | 0.10 | 0.75 | 0.76 | 0.67 | 0.69 | 0.69 | 0.57 | 0.76 | 0.76 | 0.17 | 0.91 | 0.76 | 16.45 |

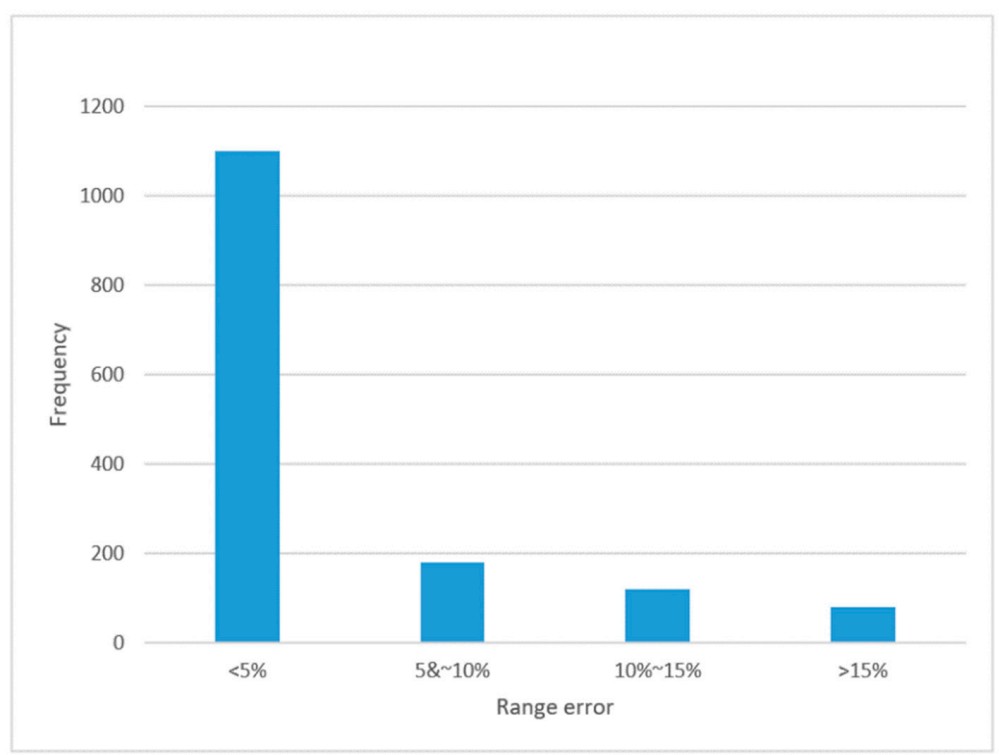

**Figure 11.** Simulation error range.

### 5.2. Analysis of the Improved Solution Method

This subsection compares the solution method employing the parallel strategy with the open-loop queueing network solution method developed by Dallery [42] with respect to the general service times. This paper improves the model developed by Dallery in the following ways:

1. It replaces the general service time with the Coxian distribution.
2. The method of decomposing complex queuing networks is applied with the parallel operation strategy.

3. The maximum entropy value method is used to calculate the open-loop queuing network.

Dallery's paper stated that the number of customers present in the network at the same time must not exceed a given value, i.e., a limited load, which is the same as the limit on the number of shuttles described in this paper. Although it does not involve parallel queuing networks, the decomposition method is similar for complex networks, and given the same parameters, can be used for comparison and validation. The method of solving the parallel operation strategy applied in this paper was compared with Dallery's method in nine environments with different mean values $\tau$ and squared coefficients of variation $cv^2$. The specific environmental parameters are shown in Table 7.

**Table 7.** Model comparison environment.

| Serial Number | Configuration | Server Node 1 | Server Node 2 |
|---|---|---|---|
| 1 | $M = 3, \lambda = 0.1$ | $(\tau, cv^2) = (3, 5)$ | $(\tau, cv^2) = (3, 5)$ |
| 2 | $M = 5, \lambda = 0.1$ | $(\tau, cv^2) = (3, 5)$ | $(\tau, cv^2) = (3, 5)$ |
| 3 | $M = 7, \lambda = 0.1$ | $(\tau, cv^2) = (3, 5)$ | $(\tau, cv^2) = (3, 5)$ |
| 4 | $M = 3, \lambda = 0.1$ | $(\tau, cv^2) = (2, 5)$ | $(\tau, cv^2) = (3, 5)$ |
| 5 | $M = 5, \lambda = 0.1$ | $(\tau, cv^2) = (2, 5)$ | $(\tau, cv^2) = (3, 5)$ |
| 6 | $M = 7, \lambda = 0.1$ | $(\tau, cv^2) = (2, 5)$ | $(\tau, cv^2) = (3, 5)$ |
| 7 | $M = 3, \lambda = 0.1$ | $(\tau, cv^2) = (1, 5)$ | $(\tau, cv^2) = (4, 5)$ |
| 8 | $M = 5, \lambda = 0.1$ | $(\tau, cv^2) = (1, 5)$ | $(\tau, cv^2) = (4, 5)$ |
| 9 | $M = 7, \lambda = 0.1$ | $(\tau, cv^2) = (1, 5)$ | $(\tau, cv^2) = (4, 5)$ |

Using the system response time $E(T)$ and the external queue waiting for pickup $L_{eq}$ as parameters for calculating the simulation results and the error of the method in this paper and that of Dallery, a comparison of the data is presented in Table 8 (the data in the table are retained to two decimal places). This paper calculates the system response time more accurately, with an average error of 3.6%, while Dallery reported an average error of 9.2%. For the external queue waiting for pickup, the method described in this paper is better overall than Dallery, with an average error of 9.46%, while the average error of Dallery is 51.9%.

**Table 8.** Comparison of model data and error rates.

| | Response Time | | | | | External Waiting Captain | | | | |
|---|---|---|---|---|---|---|---|---|---|---|
| Serial Number | Simulation | This Paper | Error (%) | Dallery | Error (%) | Simulation | This Paper | Error (%) | Dallery | Error (%) |
| 1 | 22.42 | 22.80 | 0.02 | 16.44 | 0.27 | 0.11 | 0.11 | 0.03 | 0.17 | 0.60 |
| 2 | 23.12 | 23.90 | 0.03 | 18.23 | 0.21 | 0.08 | 0.09 | 0.07 | 0.15 | 0.82 |
| 3 | 22.25 | 24.77 | 0.11 | 19.98 | 0.10 | 0.06 | 0.07 | 0.18 | 0.14 | 1.33 |
| 4 | 22.92 | 22.62 | 0.01 | 21.76 | 0.05 | 0.15 | 0.12 | 0.17 | 0.21 | 0.45 |
| 5 | 23.53 | 23.99 | 0.02 | 23.03 | 0.02 | 0.14 | 0.13 | 0.09 | 0.19 | 0.37 |
| 6 | 24.30 | 24.86 | 0.02 | 24.76 | 0.02 | 0.12 | 0.13 | 0.05 | 0.18 | 0.50 |
| 7 | 25.74 | 24.81 | 0.04 | 26.13 | 0.02 | 0.24 | 0.23 | 0.02 | 0.18 | 0.25 |
| 8 | 27.17 | 27.80 | 0.02 | 28.83 | 0.06 | 0.19 | 0.22 | 0.13 | 0.17 | 0.12 |
| 9 | 28.34 | 29.81 | 0.05 | 30.63 | 0.08 | 0.17 | 0.19 | 0.09 | 0.13 | 0.24 |

The average error for the nine environments is shown in Figure 12. Most of the error values in this paper are within 10%, with a minimum error of 1.31% and a maximum error of 18.3%, representing an overall reduction in error of 20% compared with Dallery. In summary, the methods applied both by Dallery and in this paper can be applied to solve the system performance index of the response time. However, for the external queue waiting for pickup, a comparison of the error rates between the two methods with respect to these two parameters is shown in Figure 13. The solution method described in this paper outperformed Dallery, reducing the error by approximately 40%.

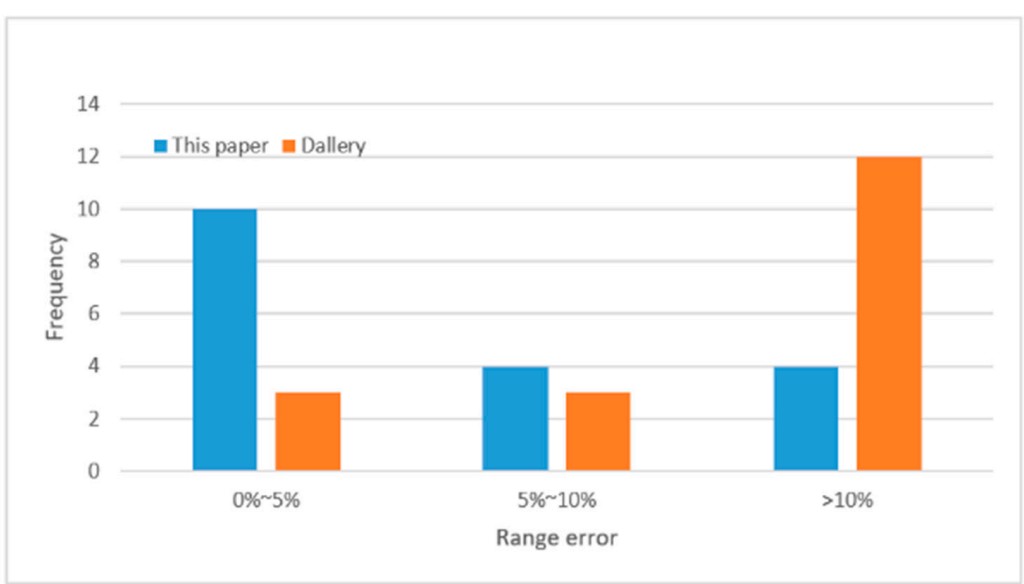

**Figure 12.** Error comparison chart.

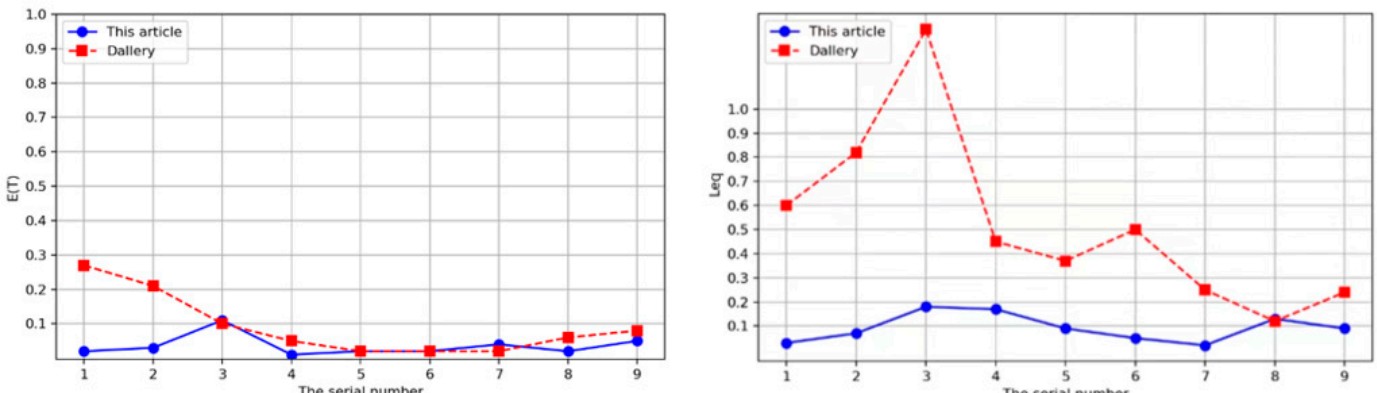

**Figure 13.** Comparison of system response time $E(T)$, external queue waiting for pair length $L_{eq}$ error under two solving methods.

### 5.3. Operation Strategy Analysis

In this paper, we mainly consider the combination of parallel and serial strategies for the four-way shuttle. Previous papers have mostly used serial operation strategies to represent the whole system, but in order to better reflect the actual system operation, parallel and serial operation strategies should be considered in combination. This subsection focuses on comparing the model built in this paper, incorporating the parallel operation strategy, with a model that uses the serial operation strategy to verify the effect of the operation strategy on the system.

To eliminate errors, 18 scales of the system were selected for validation and the number of shuttle cars in the system was set to 5 [43]; the selected system scale is shown in Table 9.

The overall response time of the serial operation strategy is longer than that of the model developed in this paper for large and small size systems with different numbers of layers and columns, increasing from arrival rate $\lambda_r = 50$ to arrival rate $\lambda_r = 250$ in 50 steps. The theoretical serial-parallel combination model established in this paper tends to increase smoothly in response time, while the single serial operation strategy has a faster growth due to the increasing arrival rate. This is because the system service time of the parallel operation strategy is the maximum value of the shuttle service time and the hoist service

time, while the system service time under the serial operation strategy is the sum of the two equipment service times.

**Table 9.** System size parameters.

| Number of Layers | **4** | | | **5** | | | **6** | | |
|---|---|---|---|---|---|---|---|---|---|
| Number of columns | 16 | 32 | 40 | 24 | 32 | 48 | 40 | 48 | 56 |
| Specification | 128 | 256 | 320 | 240 | 320 | 480 | 480 | 576 | 672 |
| Number of layers | **7** | | | **8** | | | **9** | | |
| Number of columns | 48 | 56 | 64 | 56 | 64 | 72 | 64 | 72 | 80 |
| Specification | 672 | 784 | 896 | 896 | 1024 | 1152 | 1152 | 1296 | 1440 |

In the case of a low arrival rate, the two servers, shuttle and hoist produce queues are not too long and the system response time does not result in too much time waiting for the service, so the difference between the two strategies is small. With the increasing arrival rate, the load on the hoist and shuttle increases, and there is an increase in the time that the shuttle needs to wait for the hoist when using the parallel strategy and the serial strategy. However, because the shuttle is independent of the hoist pickup when using the parallel strategy, part of the time waiting for the hoist can be spared, and the service time saved is greater than the time waiting for the hoist, while the serial strategy would cause the overall system response time to become increasingly long because of the accumulation of both the waiting time and service time.

It can be seen from Figure 14 that, at T = 4 and T = 5, there is an average of one shuttle per layer in the system, representing a fully parallel strategy; thus, the percentage reduction in the system response time is greater than that with the serial operation strategy. At T = 6, the model described in this paper, with the addition of the parallel strategy, reduces the response time by an average of 16.8% compared with the serial model, but the percentage reduction in the response time decreases with the increasing numbers of layers. At T = 9, the percentage of the performance reduction is 8.92% on average. At T = 6, T = 7, T = 8 and T = 9, a combination of 12 serial and parallel systems can reduce the system response time by 12.6% on average.

For both small-scale and large-scale systems, the model provided in this paper can reduce the response time by percentages as small as $\lambda_r = 50$, or 1.8% at T = 9. Thus, the model in this paper can improve the flexibility of the system by reducing the response time and more accurately reflecting the operation of the actual system.

### 5.4. Case Analysis

This subsection is based on a real warehouse case to verify the conclusions described in this section. The case is a four-way shuttle system for auto parts run by company A, with ten layers of shelves, 64 columns per layer, a single layer deep. The total capacity of the storage system is 1280 totes, and the current system's average daily outgoing volume is approximately 1800 totes, working 12 h a day. The other parameters of the system are shown in Table 10.

**Table 10.** System parameters.

| Parameters | Take Value | Parameters | Take Value |
|---|---|---|---|
| $Vs$ (m/s) | 2 | $\gamma_l$ (s) | 3 |
| $\varepsilon_s$ (s) | 2 | $L$ (m) | 0.65 |
| $\gamma_s$ (s) | 5 | $W$ (m) | 0.45 |
| $Vl$ (m/s) | 2.5 | $WA$ (m) | 0.45 |
| $\varepsilon_l$ (s) | 1 | $H$ (m) | 0.67 |

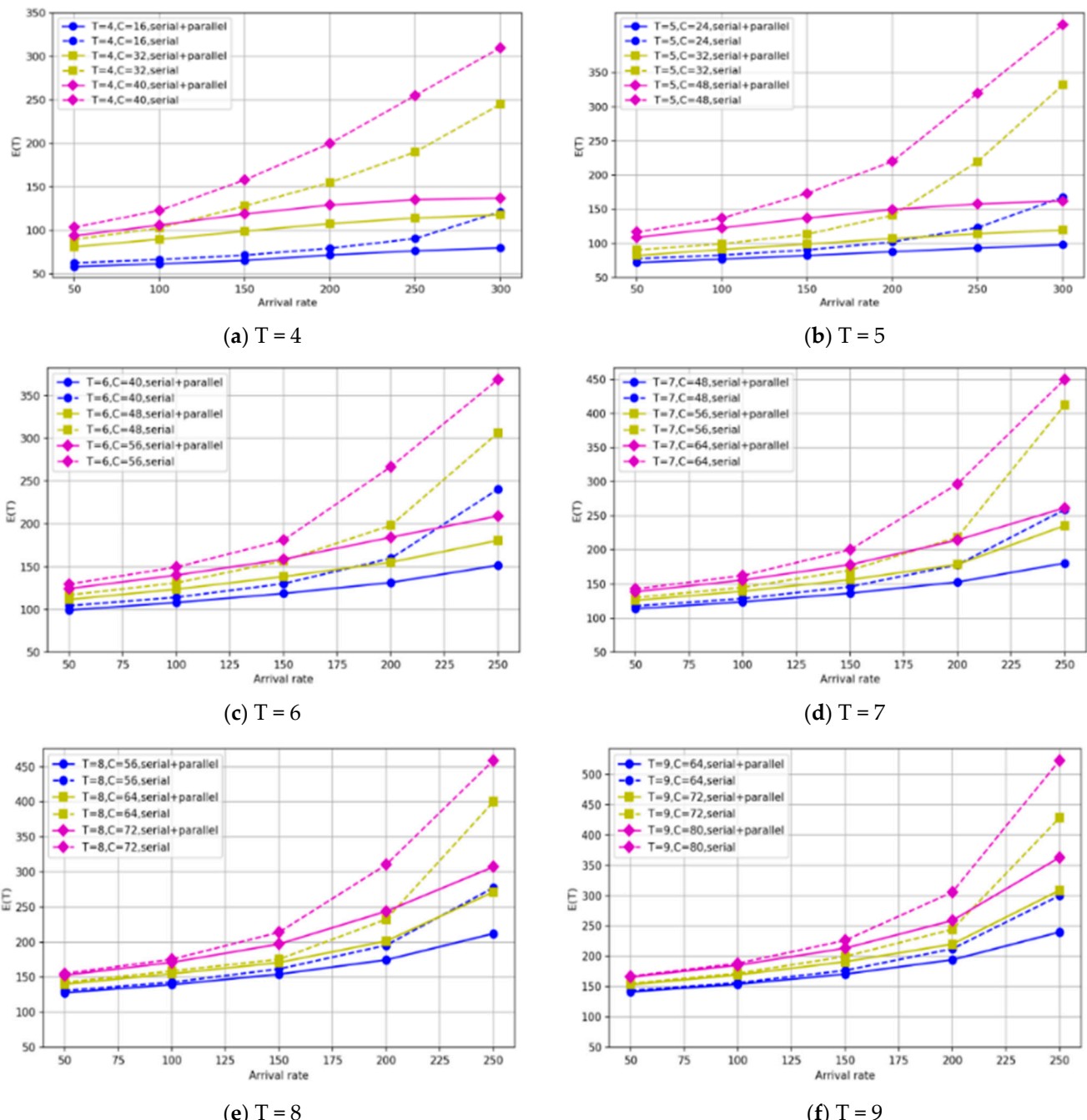

**Figure 14.** Comparison of operational policies with different system size configurations. ((**a**): The E(T) curves in the case of T = 4,C at 56, 64, and 72, respectively, for the shuttle operation strategy with a combination of serial and parallel, respectively, and the serial case alone. (**b**): The E(T) curves in the case of T = 5, C at 24, 32, and 48, respectively, for the shuttle operation strategy with a combination of serial and parallel, respectively, and the serial case alone. (**c**): The E(T) curves in the case of T = 6, C at 40, 48, and 56, respectively, for the shuttle operation strategy with a combination of serial and parallel, respectively, and the serial case alone. (**d**): The E(T) curves in the case of T = 7, C at 48, 56, and 64, respectively, for the shuttle operation strategy with a combination of serial and parallel, respectively, and the serial case alone. (**e**): The E(T) curves in the case of T = 8, C at 56, 64, and 72, respectively, for the shuttle operation strategy with a combination of serial and parallel, respectively, and the serial case alone. (**f**): The E(T) curves in the case of T = 9, C at 64, 72, and 80, respectively, for the shuttle operation strategy with a combination of serial and parallel, respectively, and the serial case alone).

The comparison between the serial operation strategy in this system and the parallel operation strategy incorporated in this paper is shown in Figure 15. Once the arrival rate reaches $\lambda_r = 160$ cases per hour, the system will increase in response to the arrival rate, which makes the number of customers served in the system tend towards the maximum load value, while the shuttles will all be in a busy state; thus, the increase in the waiting time causes the system response time to speed up. However, compared with a single serial strategy, the four-way shuttle system outbound flow model provided in this paper with the addition of a parallel strategy alleviates, to a certain extent, the excessive increase in the system response time. It can be seen from the data that when the average arrival rate of this system is 150 cases per hour, the addition of a parallel operation strategy can reduce the response time of the pickup operation by 4.17%. Between the arrival rate of 50 cases per hour and the arrival rate of 200 cases per hour, adding the parallel operation strategy reduces the response time of the picking operation by 4.36% on average.

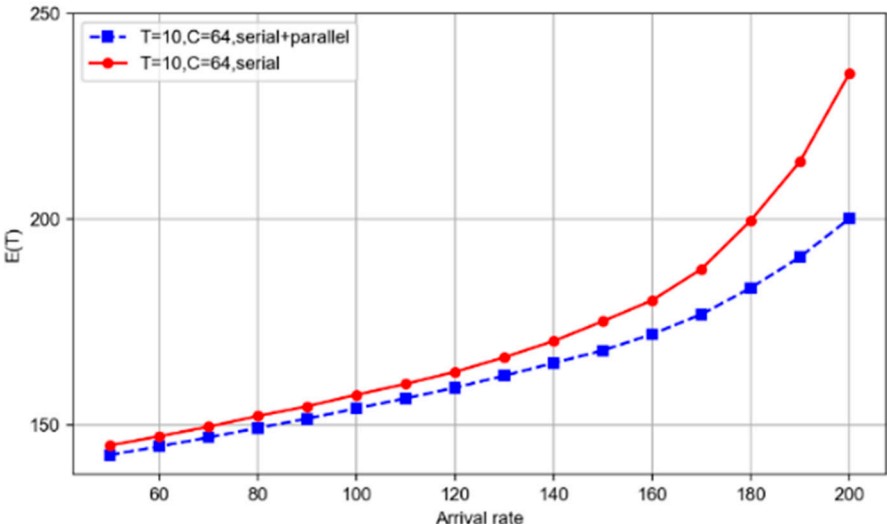

**Figure 15.** Comparison of algorithm operation strategies.

## 6. Conclusions

The main research theme of this paper is to account for parallel strategies in a four-item shuttle system, perform system modeling, system performance evaluation and system configuration studies.

In theory, the innovation in this study is the addition of the parallelism of the hoist and shuttle in the four-shuttle system, whereas scholars have typically focused on serial operation strategy, where the response time of the system is the sum of the shuttle service time and hoist service time, thus producing large errors in the calculation and evaluation of the system performance indicators. The parallel operation strategy circumvents this drawback to a certain extent.

In terms of methodology, it is considered that the hoist and shuttle service times obey a general distribution. In the modeling process of storage systems, most scholars will assume that the server service time obeys an exponential distribution, uniform distribution, etc. However, in the storage system, the service time of customers within each server is independent of each other but obeys the same distribution, and this concept is closer to the general distribution. As the general distribution does not have a fixed distribution, the queuing network formed during the solution is not easy to calculate; therefore, this paper uses the coxian distribution to approximate the solution. Arena simulation is used to verify the accuracy of the model, and the results show that the error range of simulation results is within 10% and the error of the system performance index calculation is reduced by 20% compared with the existing methods. The parallel part is integrated into a closed-loop queueing network, which preserves the characteristics of the servers within the closed-loop queueing network to a greater extent than the method used by Dallery, thus reducing the

error, and the solution method used in this paper is verified by numerical tests to reduce the error to less than 10%.

In practice, by conducting experiments and data analysis, the result is that the incorporation of a parallel operation strategy outperforms the serial operation strategy in terms of the system performance for both large-scale and small-scale systems, and this advantage rises significantly as the arrival rate increases, reducing the system response time by a minimum of 1.8% and an average of 12.6% in the numerical experiments. For the selection of the number of shuttles to be studied. the four-way shuttle system will stabilise at 80% shuttle utilisation and will not exceed the maximum system load (number of shuttles). After the shuttle utilisation is greater than 80%, the system response time and external queue waiting time will increase significantly, thus allowing 80% shuttle utilisation to be used as the optimal number of shuttles in the system at different arrival rates for the node configuration.

Shortcomings and Prospects

In this paper, the serial and parallel operation strategies are probabilistically distributed, and the "decide" module is used in the simulation. However, in real situations, there are many factors that influence the operation strategy, which are difficult to be covered by the probabilistic approach alone. The current research on automatic storage systems assumes a random storage strategy, but the existing sorting and positioning storage strategies are conducive to improving the system throughput and operational efficiency. Due to the complex distribution of aisles in the four-way shuttle system, the shuttles inevitably encounter conflicts between the shuttles in the access process. Therefore, these factors will be in the focus of future research.

**Author Contributions:** Conceptualization, J.M. and J.C.; methodology, J.C.; software, J.C.; validation, X.L. and H.Z.; formal analysis, J.C. and C.L.; investigation, J.M.; resources, X.L.; data curation, J.C.; writing—original draft preparation, J.C.; writing—review and editing, H.Z.; visualization, C.L. and X.L.; supervision, C.L. and H.Z.; project administration, H.Z. and C.L. All authors have read and agreed to the published version of the manuscript.

**Funding:** This research received no external funding.

**Institutional Review Board Statement:** Not applicable.

**Informed Consent Statement:** Not applicable.

**Data Availability Statement:** Not applicable.

**Conflicts of Interest:** The authors declare no conflict of interest.

## Abbreviations

| | |
|---|---|
| OQN | Open-loop queuing network |
| CQN | Closed-loop queuing network |
| SOQN | Semi-open-loop queuing network |
| NQN | Nested queuing network |
| AVS/RS | Automated vehicle storage and retrieval system |
| AS/RS | Automated storage and retrieval system |
| SBS/RS | Shuttle based storage and retrieval system |
| MGM | Matrix geometric method |
| FCFS | First-come, first-served |
| PH | Phase-type |
| SKU | Stock keeping unit |
| WMS | Warehouse management system |
| MEM | Maximum entropy method |
| R/S | Storage/Retrieval |

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
