# Peer review of "Modelling Analysis of a Four-Way Shuttle-Based Storage and Retrieval System on the Basis of Operation Strategy"

_applsci, doi:10.3390/app13053306_

Round 1
Reviewer 1 Report
The article addresses the issues of system modeling, system performance evaluation and system configuration studies for a four element shuttle system that includes a parallel operation strategy. The authors showed the advantages of this method. The authors approached the subject comprehensively: in the scientific sense. Features of the article to be highlighted: Advanced math apparatus; Well prepared and interesting illustrative material;
Very useful summaries of information in tables.
Additional review comments:
1. This article focuses on the following question:
System modeling, system performance evaluation, and system configuration studies for a four-element shuttle system that includes a parallel operation strategy.
2. The topic is not new, but the authors' approach to the issue is original. The authors propose a new way of solving the problem - the strategy of parallel system operation in automatic storage systems. The authors approached the subject professionally in the scientific sense.
3. The work introduces a new way of solving the problem - the strategy of parallel system operation in automatic storage systems. This is a different, original approach to the subject.
4. The authors should continue modeling in this direction. They should strive for experimental verification of their solutions on real objects. It would be an interesting sequel, and the observations and conclusions would certainly be interesting.
5. The conclusions are consistent with the evidence and arguments presented. They answer specifically the main question posed.
6. In my opinion, the references are adequate, sufficient.
7. Advanced math apparatus. Well prepared and interesting illustrative material. Very useful summaries of information in tables.
Author Response
Response to Reviewer 1 Comments
Point 1: This article focuses on the following question:
System modeling, system performance evaluation, and system configuration studies for a four-element shuttle system that includes a parallel operation strategy.
Response 1: Thank you very much for your comments.
Point 2: The topic is not new, but the authors' approach to the issue is original. The authors propose a new way of solving the problem - the strategy of parallel system operation in automatic storage systems. The authors approached the subject professionally in the scientific sense.
Response 2: Thank you very much for your comments.
Point 3: The work introduces a new way of solving the problem - the strategy of parallel system operation in automatic storage systems. This is a different, original approach to the subject.
Response 3: Thank you very much for your comments.
Point 4: The authors should continue modeling in this direction. They should strive for experimental verification of their solutions on real objects. It would be an interesting sequel, and the observations and conclusions would certainly be interesting.
Response 4: Thank you very much for your suggestions. In future research, we will use this as the basis for more complex and in-depth model and algorithm building. Future research will be studied with practical applications and experiments will be built for validation.
Point 5: The conclusions are consistent with the evidence and arguments presented. They answer specifically the main question posed.
Response 5: Thank you very much for your comments.
Point 6: In my opinion, the references are adequate, sufficient.
Response 6: Thank you very much for your comments.
Point 7: Advanced math apparatus. Well prepared and interesting illustrative material. Very useful summaries of information in tables.
Response 7: Thank you very much for your comments.

Reviewer 2 Report
The paper rationalizes the outbound process of the four-way shuttle system by focusing on the modeling, performance evaluation and configuration of the four-way shuttle system based on parallel operation strategy to reduce resource waste achieving sustainable development. This topic is interesting and actual
I consider the topic original and relent in the field. The paper is innovative
Paper in general is good. My concern is about the analysis or results. Tables are dificult to follow and understand (for example, table 8). They need more explanations and perhaps other more clear format. Also theoretical background must be extended. Instead of the results, the theory in which are based must be explained in more detail.
The conclusions are good and well explained. I consider that references are apprpriate
About the results. Tables are dificult to follow and understand (for example, table 8). They need more explanations and perhaps other more clear format.
Author Response
Response to Reviewer 2 Comments
Point 1: Paper in general is good. My concern is about the analysis or results. Tables are dificult to follow and understand (for example, table 8). They need more explanations and perhaps other more clear format. Also theoretical background must be extended. Instead of the results, the theory in which are based must be explained in more detail.
Response 1: Thank you very much for your comments.
Point 2: The conclusions are good and well explained. I consider that references are apprpriate
Point 3: About the results. Tables are dificult to follow and understand (for example, table 8). They need more explanations and perhaps other more clear format.
Response 2, 3: Thank you very much for your comments. In response to your question about your concern that the analysis or results are difficult to read, We give the following explanations here. Based on the fact that the four-way shuttle system size will vary according to the size of the business volume, this paper lists seven different four-way shuttle system sizes with a different number of levels and columns as well as setting five different arrival rates, with the arrival rate set at per hour. The metrics calculated in the simulation are system response time, external queue waiting captain, shuttle utilization, and the error values of simulation results and model-solving results. Numbers 1-6 refer to a total of six experiments done to obtain the metrics at each different arrival rate. The table has been adjusted for a clearer description. Hope to bring a clearer format.

Reviewer 3 Report
Paper is well written. can be accepted in the current form.
Author Response
Response to Reviewer 3 Comments
Point 1: Paper is well written. can be accepted in the current form.
Response 1: Thank you very much for your comments.

Reviewer 4 Report
Dear Authors,
For a review comments, please see the Attachment.
Sincerely,
Reviewer

Author Response
Response to Reviewer 4 Comments
General comment reply
1)Managerial implications and insights are missing from the paper. Please emphasize the applicability of your model in a real-life setting. Who would use your model? What benefits will it bring? Why your model and not others in the literature? What do your results recommend managers do? How much costsavings would your model bring? How easy is it to implement it in practice? Are there any obstacles? Limitations?
Response:Traditional automated storage system equipment covers a large area, low operational efficiency and poor flexibility. Because the four-way shuttle can be completed in the system before, after, left, right, up and down the reciprocal movement, so it appears to improve the performance of the storage system, circumventing the defects of the traditional automated warehouse. Based on this, the four-way shuttle system is the more advanced, highly automated, highly flexible dense storage system in existence.
Four-way shuttle system can be realized by four-way shuttle any number of layers, any aisle movement, and horizontal movement does not require a hoist to assist in changing lanes, which makes the system has a strong flexibility, but this involves the shuttle and hoist operation strategy issues. The system studied in this paper can be configured with different numbers of shuttles according to the size of the actual business volume in real-life enterprises to achieve flexible planning with optimal efficiency and lowest cost. Therefore, this paper takes the four-way shuttle system as the research object, and focuses on the system model establishment, system performance evaluation and system configuration problems by adding parallel operation strategies to achieve the purpose of improving the overall system efficiency and equipment operation efficiency.
For enterprises, cost reduction and efficiency improvement is one of the key factors that can improve their competitiveness in the competitive market. Four-way shuttle system can provide enterprises with high flexibility, high efficiency, high throughput rate, low cost storage system environment, and can add or remove the number of shuttles to meet different business needs, but how to choose the appropriate number of shuttles for different needs has become an urgent problem for enterprises to solve. In this paper, we establish a more realistic theoretical model, establish, calculate and analyze the system performance indicators, study the impact of parameter changes on the system efficiency and determine the optimal number of shuttles to meet the throughput according to different needs to avoid wasting resources.
This paper mainly lacks research on the congestion and blockage of shuttles in the system. In actual life, in enterprise applications, due to the complex distribution of lanes in the four-way shuttle system, shuttles inevitably encounter conflicts between shuttles in the access process, based on which the problems of shuttle path optimization and order sequence optimization of outgoing orders are worth studying.
2)Please justify your assumption of the parameters. Comparing your model with other prominent ones in the literature would bring additional managerial insights and value.
Response:Thank you very much for your suggestion. The research hypothesis of this paper is based on the hypothesis building based on the reference of previous research literature by scholars, based on the analysis of those factors that have an impact on us after the research and the needs of the actual situation. In the simulation analysis part of the article, we compare our results with previous scholars' studies to verify the accuracy of our model. In the literature review section, we also illustrate the differences and connections between the literature through tables as well as in the form of textual representations.
Point 2: Key words need to be revised as follows: automated warehouses, four-way shuttle systems, queuing network model, analytical and numerical modelling, performance analysis.
Response 2: Thank you very much for your suggestion. We have taken your suggestion and will modify the keywords as you proposed.
Point 3: Introduction → I encourage the authors to discuss how their research can be used in practice, give real-life examples of companies who needed the proposed research to increase the system performance of their warehouse systems.
Response 3: Thank you very much for your suggestion. During the simulation experiment, the authors set different parameters for the simulation. The aim is to test the accuracy of the model as much as possible while meeting the needs of different companies. However, in this paper, the results are analyzed in Chapter 6 simulation experiments with an example of a researched company.
Point 4: Introduction → along with Figure 1 additional figure of the shuttle with a shelf should be
presented (see below).
Response 4: Thank you very much for your suggestion, we have added pictures to the paper.
Point 5: Introduction → the authors should use the abbreviations that are accepted in the research community, as follows: Shuttle based Storage and Retrieval System → SBS/RS − Autonomous Vehicles Storage and Retrieval Systems →AVS/RS − Speed → velocity.
Response 5: Thank you very much for your suggestion. We have added a complete list of abbreviations for the entire article in the appendix.
Point 6: Literature review → I encourage the authors to extend their literature review by addressing some additional papers from classical Shuttle based Storage and Retrieval System (SBS/RS) and Autonomous Vehicles Storage and Retrieval Systems (AVS/RS). Analytical difficulties and differences of the proposed research compared to already published papers should be addressed well.
Response 6: Your suggestion is greatly appreciated. Our article contains AVS/RS literatures and compares the literatures with the studies in this article in the form of tables and textual representations. We have added SBS/RS subject literature to the article as you suggested.
Point 7: Assumptions → authors should present more focused assumptions that have been used in their research study.
Point 13: Travel time model → the authors should comment the assumption on which shuttles are operating (v = const. or v ≠ const.). The authors should comment why they have decided for such an assumption. Please explain how this assumption will affect the correctness of your analytical
model.
Response 7, 13: Thank you very much for your suggestion, and in response to your question about the modeling assumptions in this paper, our response is that the focus of this paper is on the modeling, performance evaluation, and configuration of a four-way shuttle system based on a parallel operation strategy. Firstly, the object of study in this paper is a four-way shuttle system with one hoist and the number of aisles it serves, and the shuttles it serves as an overall multi-layered, multi-aisle unit to map the performance of the whole system. The system uses one tote as the access unit, and only one tote can be accessed per bay, and the bays are the same size. For the convenience of the study and without loss of generality, the research hypothesis is based on a hypothesis building process based on references to previous research literature by scholars, after research and the needs of the actual situation, by analyzing those factors that have an impact on us.
Point 8: Abbreviations → a list of all abbreviations is missing.
Response 8: We have added a complete table of abbreviations in the appendix.
Point 9: Symbols → a list of all symbols is missing.
Point 10: Symbols → the authors are encouraged to go carefully once again through all symbols that are listed in the paper in order to give a complete list of symbols.
Point 12: Analytical model → analytical model is relatively hard to read and to understand. Because of the
lack of the list of all symbols, I could not prove the analytical model.
Response 9, 10, 12: Thank you very much for your suggestion. We have checked all the symbols in the article and made a complete list of them.
Point 11: All symbols in the text needs to be written in the italic written style. I propose the authors to check all the pages and correct the symbols to the italic written style.
Response 11: Thank you very much for your suggestion. We have checked all the symbols and changed them to italics.
Point 14: Figures have poor resolution and need to be saved and submit in the postscript format such as eps., or wmf., or in some other similar format, which gives the highest resolution when printing the paper.
Response 14:For the unclear images, we have replaced them.
Point 15: Numerical study → there is not enough information’s for the numerical study. The authors should
give more information’s about the input data for the analysis, so that the analysis and results could be repeatable. I propose that the authors use an algorithm in order to present their main logic of the numerical model.
Response 15: Thank you very much for your suggestion. For what you have suggested there is not enough information for numerical studies. There is a lot of data to be processed in our articles and especially in simulation experiments. In our future research, we collect more data for processing and apply more sophisticated algorithms to make our results more convincing and of more general significance.
Point 16: Analysis → the authors should analyse the throughput performance according to several variables that correspond to shuttle vehicles and the warehouse. I am not sure if the current analysis gives enough information to warehouse designers and to warehouse managers for the application in practice. I encourage the authors to include in the throughput performance as number Sku’s per hour.
Response 16: Thank you very much for your suggestion. Due to time constraints, we can base our future research on the current study on the number of Sku per hour in the performance of throughput, with the aim of being able to provide warehouse designers and warehouse managers with enough information to apply in practice.
Point 17: Analysis → the authors should present the most important results in graphs. In this way the warehouse managers and warehouse integrators could have a quick and clear vision on most important results.
Response 17: Thank you very much for your suggestion. In response to your question that "the authors should graph the most important results", clear tables of data and graphs of results are given in the research section as well as in the simulation section.
Point 18: Analysis → needs to be modified with more in-depth discussions of results presented in tables. Only
observations from the results are presented, however few comments or remarks are given as to why such results are obtained. The authors are encouraged to give more details about the reason why these observations are happened.
Response 18: Thank you very much for your suggestion. For this issue you raised, the results are discussed in detail in Chapter 6 of this paper, based on the simulation data, and presented in the form of pictures and tables. The results of this paper are also compared with the previous studies by scholars. It is also discussed and verified with real warehouse case example.
Point 19: Conclusions needs to be extended with more in-depth discussions of the results and with the future work.
Response 19: Thank you very much for your suggestion, and after careful study, we have revised our conclusion.
Point 20: There are some typos in the text.
Response 20: Thank you very much for your suggestion. We apologize for some language errors in the article. We have applied the language editing service provided in MDPI in order to improve the language expressions. We hope this will improve the language of our paper.
Point 21: General remark → I am not sure if the reference style is correct and in accordance with the Scientific reports. I encourage the authors to use the reference style from Scientific reports and to follow this style correctly.
Response 21: Thank you very much for your suggestion. The references for our paper follow the MDPI's reference format.

Round 2
Reviewer 4 Report
/